# Rethinking Bias Mitigation: Fairer Architectures Make for Fairer Face Recognition

**Samuel Dooley**[*]
University of Maryland, Abacus.AI
samuel@abacus.ai

**Rhea Sanjay Sukthanker**[*]
University of Freiburg
sukthank@cs.uni-freiburg.de

**John P. Dickerson**
University of Maryland, Arthur AI
johnd@umd.edu

**Colin White**
Caltech, Abacus.AI
crwhite@caltech.edu

**Frank Hutter**
University of Freiburg
fh@cs.uni-freiburg.de

**Micah Goldblum**
New York University
goldblum@nyu.edu

## Abstract

Face recognition systems are widely deployed in safety-critical applications, including law enforcement, yet they exhibit bias across a range of socio-demographic dimensions, such as gender and race. Conventional wisdom dictates that model biases arise from biased training data. As a consequence, previous works on bias mitigation largely focused on pre-processing the training data, adding penalties to prevent bias from effecting the model during training, or post-processing predictions to debias them, yet these approaches have shown limited success on hard problems such as face recognition. In our work, we discover that biases are actually inherent to neural network architectures themselves. Following this reframing, we conduct the first neural architecture search for fairness, jointly with a search for hyperparameters. Our search outputs a suite of models which Pareto-dominate all other high-performance architectures and existing bias mitigation methods in terms of accuracy and fairness, often by large margins, on the two most widely used datasets for face identification, CelebA and VGGFace2. Furthermore, these models generalize to other datasets and sensitive attributes. We release our code, models and raw data files at https://github.com/dooleys/FR-NAS.

## 1 Introduction

Machine learning is applied to a wide variety of socially-consequential domains, e.g., credit scoring, fraud detection, hiring decisions, criminal recidivism, loan repayment, and face recognition [78, 81, 61, 3], with many of these applications significantly impacting people's lives, often in discriminatory ways [5, 55, 114]. Dozens of formal definitions of fairness have been proposed [80], and many algorithmic techniques have been developed for debiasing according to these definitions [106]. Existing debiasing algorithms broadly fit into three (or arguably four [96]) categories: pre-processing [e.g., 32, 93, 89, 110], in-processing [e.g., 123, 124, 25, 35, 83, 110, 73, 79, 24, 59], or post-processing [e.g., 44, 114].

---

[*] indicates equal contribution

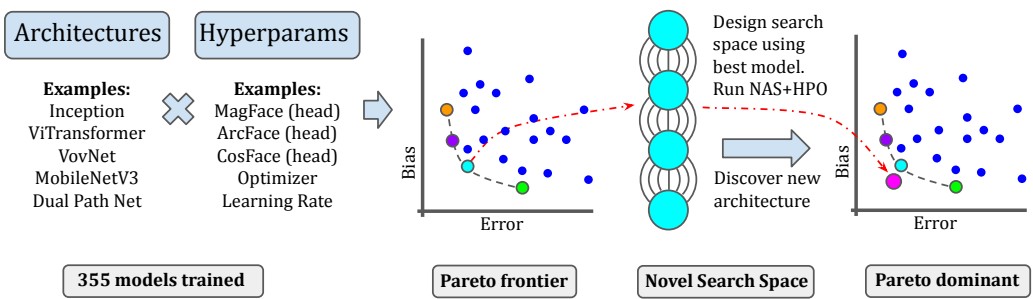

Figure 1: Overview of our methodology.

Conventional wisdom is that in order to effectively mitigate bias, we should start by selecting a model architecture and set of hyperparameters which are optimal in terms of accuracy and then apply a mitigation strategy to reduce bias. This strategy has yielded little success in hard problems such as face recognition [14]. Moreover, even randomly initialized face recognition models exhibit bias and in the same ways and extents as trained models, indicating that these biases are baked in to the architectures already [13]. While existing methods for debiasing machine learning systems use a fixed neural architecture and hyperparameter setting, we instead, ask a fundamental question which has received little attention: *Does model bias arise from the architecture and hyperparameters?* Following an affirmative answer to this question, we exploit advances in neural architecture search (NAS) [30] and hyperparameter optimization (HPO) [33] to search for inherently fair models.

We demonstrate our results on face identification systems where pre-, post-, and in-processing techniques have fallen short of debiasing face recognition systems. Training fair models in this setting demands addressing several technical challenges [14]. Face identification is a type of face recognition deployed worldwide by government agencies for tasks including surveillance, employment, and housing decisions. Face recognition systems exhibit disparity in accuracy based on race and gender [37, 92, 91, 61]. For example, some face recognition models are 10 to 100 times more likely to give false positives for Black or Asian people, compared to white people [2]. This bias has already led to multiple false arrests and jail time for innocent Black men in the USA [48].

In this work, we begin by conducting the first large-scale analysis of the impact of architectures and hyperparameters on bias. We train a diverse set of 29 architectures, ranging from ResNets [47] to vision transformers [28, 68] to Gluon Inception V3 [103] to MobileNetV3 [50] on the two most widely used datasets in face identification that have socio-demographic labels: CelebA [69] and VGGFace2 [8]. In doing so, we discover that architectures and hyperparameters have a significant impact on fairness, across fairness definitions.

Motivated by this discovery, we design architectures that are simultaneously fair and accurate. To this end, we initiate the study of NAS for fairness by conducting the first use of NAS+HPO to jointly optimize fairness and accuracy. We construct a search space informed by the highest-performing architecture from our large-scale analysis, and we adapt the existing Sequential Model-based Algorithm Configuration method (SMAC) [66] for multi-objective architecture and hyperparameter search. We discover a Pareto frontier of face recognition models that outperform existing state-of-the-art models on both test accuracy and multiple fairness metrics, often by large margins. An outline of our methodology can be found in Figure 1.

We summarize our primary contributions below:

- By conducting an exhaustive evaluation of architectures and hyperparameters, we uncover their strong influence on fairness. Bias is inherent to a model's inductive bias, leading to a substantial difference in fairness across different architectures. We conclude that the implicit convention of choosing standard architectures designed for high accuracy is a losing strategy for fairness.
- Inspired by these findings, we propose a new way to mitigate biases. We build an architecture and hyperparameter search space, and we apply existing tools from NAS and HPO to automatically design a fair face recognition system.
- Our approach finds architectures which are Pareto-optimal on a variety of fairness metrics on both CelebA and VGGFace2. Moreover, our approach is Pareto-optimal compared to other previous bias mitigation techniques, finding the fairest model.

- The architectures we synthesize via NAS and HPO generalize to other datasets and sensitive attributes. Notably, these architectures also reduce the linear separability of protected attributes, indicating their effectiveness in mitigating bias across different contexts.

We release our code and raw results at https://github.com/dooleys/FR-NAS, so that users can easily adapt our approach to any bias metric or dataset.

## 2 Background and Related Work

**Face Identification.**    Face recognition tasks can be broadly categorized into two distinct categories: *verification* and *identification*. Our specific focus lies in face *identification* tasks which ask whether a given person in a source image appears within a gallery composed of many target identities and their associated images; this is a one-to-many comparison. Novel techniques in face recognition tasks, such as ArcFace [108], CosFace [23], and MagFace [75], use deep networks (often called the *backbone*) to extract feature representations of faces and then compare those to match individuals (with mechanisms called the *head*). Generally, *backbones* take the form of image feature extractors and *heads* resemble MLPs with specialized loss functions. Often, the term "head" refers to both the last layer of the network and the loss function. Our analysis primarily centers around the face identification task, and we focus our evaluation on examining how close images of similar identities are in the feature space of trained models, since the technology relies on this feature representation to differentiate individuals. An overview of these topics can be found in Wang and Deng [109].

**Bias Mitigation in Face Recognition.**    The existence of differential performance of face recognition on population groups and subgroups has been explored in a variety of settings. Earlier work [e.g., 57, 82] focuses on single-demographic effects (specifically, race and gender) in pre-deep-learning face detection and recognition. Buolamwini and Gebru [5] uncover unequal performance at the phenotypic subgroup level in, specifically, a gender classification task powered by commercial systems. Raji and Buolamwini [90] provide a follow-up analysis – exploring the impact of the public disclosures of Buolamwini and Gebru [5] – where they discovered that named companies (IBM, Microsoft, and Megvii) updated their APIs within a year to address some concerns that had surfaced. Further research continues to show that commercial face recognition systems still have socio-demographic disparities in many complex and pernicious ways [29, 27, 54, 54, 26].

Facial recognition is a large and complex space with many different individual technologies, some with bias mitigation strategies designed just for them [63, 118]. The main bias mitigation strategies for facial identification are described in Section 4.2.

**Neural Architecture Search (NAS) and Hyperparameter Optimization (HPO).**    Deep learning derives its success from the manually designed feature extractors which automate the feature engineering process. Neural Architecture Search (NAS) [30, 116], on the other hand, aims at automating the very design of network architectures for a task at hand. NAS can be seen as a subset of HPO [33], which refers to the automated search for optimal hyperparameters, such as learning rate, batch size, dropout, loss function, optimizer, and architectural choices. Rapid and extensive research on NAS for image classification and object detection has been witnessed as of late [67, 125, 121, 88, 6]. Deploying NAS techniques in face recognition systems has also seen a growing interest [129, 113]. For example, reinforcement learning-based NAS strategies [121] and one-shot NAS methods [113] have been deployed to search for an efficient architecture for face recognition with low *error*. However, in a majority of these methods, the training hyperparameters for the architectures are *fixed*. We observe that this practice should be reconsidered in order to obtain the fairest possible face recognition systems. Moreover, one-shot NAS methods have also been applied for multi-objective optimization [39, 7], e.g., optimizing accuracy and parameter size. However, none of these methods can be applied for a joint architecture and hyperparameter search, and none of them have been used to optimize *fairness*.

For the case of tabular datasets, a few works have applied hyperparameter optimization to mitigate bias in models. Perrone et al. [87] introduced a Bayesian optimization framework to optimize accuracy of models while satisfying a bias constraint. Schmucker et al. [97] and Cruz et al. [17] extended Hyperband [64] to the multi-objective setting and showed its applications to fairness. Lin et al. [65] proposed de-biasing face recognition models through model pruning. However, they only considered two architectures and just one set of fixed hyperparameters. To the best of our knowledge,

no prior work uses any AutoML technique (NAS, HPO, or joint NAS and HPO) to design fair face recognition models, and no prior work uses NAS to design fair models for any application.

# 3    Are Architectures and Hyperparameters Important for Fairness?

In this section, we study the question *"Are architectures and hyperparameters important for fairness?"* and report an extensive exploration of the effect of model architectures and hyperparameters.

**Experimental Setup.**    We train and evaluate each model configuration on a gender-balanced subset of the two most popular face identification datasets: CelebA and VGGFace2. CelebA [69] is a large-scale face attributes dataset with more than 200K celebrity images and a total of 10 177 gender-labeled identities. VGGFace2 [8] is a much larger dataset designed specifically for face identification and comprises over 3.1 million images and a total of 9 131 gender-labeled identities. While this work analyzes phenotypic metadata (perceived gender), the reader should not interpret our findings absent a social lens of what these demographic groups mean inside society. We guide the reader to Hamidi et al. [40] and Keyes [56] for a look at these concepts for gender.

To study the importance of architectures and hyperparameters for fairness, we use the following training pipeline – ultimately conducting 355 training runs with different combinations of 29 architectures from the Pytorch Image Model (`timm`) database [117] and hyperparameters. For each model, we use the default learning rate and optimizer that was published with that model. We then train the model with these hyperparameters for each of three heads, ArcFace [108], CosFace [23], and MagFace [75]. Next, we use the model's default learning rate with both AdamW [70] and SGD optimizers (again with each head choice). Finally, we also train with AdamW and SGD with unified learning rates (SGD with `learning_rate`=0.1 and AdamW with `learning_rate`=0.001). In total, we thus evaluate a single architecture between 9 and 13 times (9 times if the default optimizer and learning rates are the same as the standardized, and 13 times otherwise). All other hyperparameters are held constant fortraining of the model.

**Evaluation procedure.**    As is commonplace in face identification tasks [12, 13], we evaluate the performance of the learned representations. Recall that face recognition models usually learn representations with an image backbone and then learn a mapping from those representations onto identities of individuals with the head of the model. We pass each test image through a trained model and save the learned representation. To compute the representation error (which we will henceforth simply refer to as *Error*), we merely ask, for a given probe image/identity, whether the closest image in feature space is *not* of the same person based on $l_2$ distance. We split each dataset into train, validation, and test sets. We conduct our search for novel architectures using the train and validation splits, and then show the improvement of our model on the test set.

The most widely used fairness metric in face identification is *rank disparity*, which is explored in the NIST FRVT [38]. To compute the rank of a given image/identity, we ask how many images of a different identity are closer to the image in feature space. We define this index as the rank of a given image under consideration. Thus, Rank(image) = 0 if and only if Error(image) = 0; Rank(image) > 0 if and only if Error(image) = 1. We examine the **rank disparity**: the absolute difference of the average ranks for each perceived gender in a dataset $\mathcal{D}$:

$$\left| \frac{1}{|\mathcal{D}_{\text{male}}|} \sum_{x \in \mathcal{D}_{\text{male}}} \text{Rank}\left(x\right) - \frac{1}{|\mathcal{D}_{\text{female}}|} \sum_{x \in \mathcal{D}_{\text{female}}} \text{Rank}(x) \right|. \tag{1}$$

We focus on rank disparity throughout the main body of this paper as it is the most widely used in face identification, but we explore other forms of fairness metrics in face recognition in Appendix C.4.

**Results and Discussion.**    By plotting the performance of each training run on the validation set with the error on the $x$-axis and rank disparity on the $y$-axis in Figure 2, we can easily conclude two main points. First, optimizing for error does not always optimize for fairness, and second, different architectures have different fairness properties. We also find the DPN architecture has the lowest error and is Pareto-optimal on both datasets; hence, we use that architecture to design our search space in Section 4.

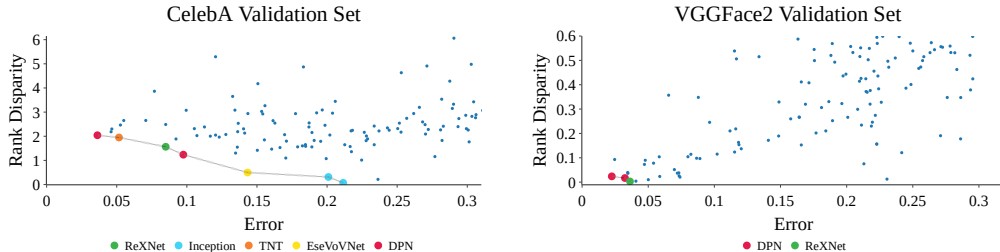

Figure 2: (Left) CelebA (Right) VGGFace2. Error-Rank Disparity Pareto front of the architectures with lowest error (< 0.3). Models in the lower left corner are better. The Pareto front is denoted with a dashed line. Other points are architecture and hyperparameter combinations which are not Pareto-optimal.

We note that in general there is a low correlation between error and rank disparity (e.g., for models with error < 0.3, $\rho = .113$ for CelebA and $\rho = .291$ for VGGFace2). However, there are differences between the two datasets at the most extreme low errors. First, for VGGFace2, the baseline models already have very low error, with there being 10 models with error < 0.05; CelebA only has three such models. Additionally, models with low error also have low rank disparity on VGGFace2 but this is not the case for CelebA. This can be seen by looking at the Pareto curves in Figure 2.

The Pareto-optimal models also differ across datasets: on CelebA, they are versions of DPN, TNT, ReXNet, VovNet, and ResNets, whereas on VGGFace2 they are DPN and ReXNet. Finally, we note that different architectures exhibit different optimal hyperparameters. For example, on CelebA, for the Xception65 architecture finds the combinations of (SGD, ArcFace) and (AdamW, ArcFace) as Pareto-optimal, whereas the Inception-ResNet architecture finds the combinations (SGD, MagFace) and (SGD, CosFace) Pareto-optimal.

## 4 Neural Architecture Search for Bias Mitigation

Inspired by our findings on the importance of architecture and hyperparameters for fairness in Section 3, we now initiate the first joint study of NAS for fairness in face recognition, also simultaneously optimizing hyperparameters. We start by describing our search space and search strategy. We then compare the results of our NAS+HPO-based bias mitigation strategy against other popular face recognition bias mitigation strategies. We conclude that our strategy indeed discovers simultaneously accurate and fair architectures.

### 4.1 Search Space Design and Search Strategy

We design our search space based on our analysis in Section 3, specifically around the Dual Path Networks[10] architecture which has the lowest error and is Pareto-optimal on both datasets, yielding the best trade-off between rank disparity and accuracy as seen in Figure 2.

**Hyperparameter Search Space Design.** We optimize two categorical hyperparameters (the architecture head/loss and the optimizer) and one continuous one (the learning rate). The learning rate's range is conditional on the choice of optimizer; the exact ranges are listed in Table 6 in the appendix.

**Architecture Search Space Design.** Dual Path Networks [10] for image classification share common features (like ResNets [46]) while possessing the flexibility to explore new features [52] through a dual path architecture. We replace the repeating `1x1_conv-3x3_conv-1x1_conv` block with a simple recurring searchable block. Furthermore, we stack multiple such searched blocks to closely follow the architecture of Dual Path Networks. We have nine possible choices for each of the three operations in the DPN block, each of which we give a number 0 through 8. The choices include a vanilla convolution, a convolution with pre-normalization and a convolution with post-normalization, each of them paired with kernel sizes 1×1, 3×3, or 5×5 (see Appendix C.2 for full details). We thus have 729 possible architectures (in addition to an infinite number of hyperparameter configurations). We denote each of these architectures by XYZ where $X, Y, Z \in \{0, \ldots, 8\}$; e.g., architecture 180 represents the architecture which has operation 1, followed by operation 8, followed by operation 0.

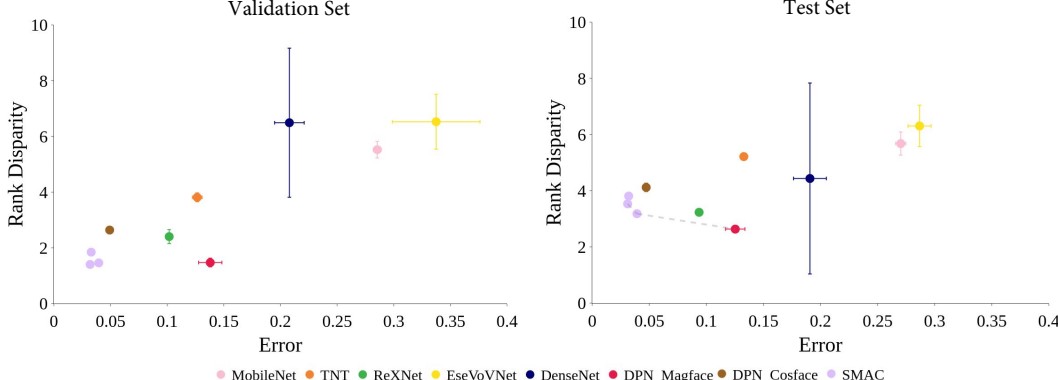

Figure 3: Pareto front of the models discovered by SMAC and the rank-1 models from `timm` for the *(a)* validation and *(b)* test sets on CelebA. Each point corresponds to the mean and standard error of an architecture after training for 3 seeds. The SMAC models Pareto-dominate the top performing `timm` models ($Error < 0.1$).

**Search strategy.** To navigate this search space we have the following desiderata:

- **Joint NAS+HPO.** Since there are interaction effects between architectures and hyperparameters, we require an approach that can jointly optimize both of these.

- **Multi-objective optimization.** We want to explore the trade-off between the accuracy of the face recognition system and the fairness objective of choice, so our joint NAS+HPO algorithm needs to supports multi-objective optimization [84, 21, 71].

- **Efficiency.** A single function evaluation for our problem corresponds to training a deep neural network on a given dataset. As this can be quite expensive on large datasets, we would like to use cheaper approximations with multi-fidelity optimization techniques [98, 64, 31].

To satisfy these desiderata, we employ the multi-fidelity Bayesian optimization method SMAC3 [66] (using the SMAC4MF facade), casting architectural choices as additional hyperparameters. We choose Hyperband [64] for cheaper approximations with the initial and maximum fidelities set to 25 and 100 epochs, respectively, and $\eta = 2$. Every architecture-hyperparameter configuration evaluation is trained using the same training pipeline as in Section 3. For multi-objective optimization, we use the ParEGO [21] algorithm with $\rho$ set to 0.05.

## 4.2 Empirical Evaluation

We now report the results of our NAS+HPO-based bias mitigation strategy. First, we discuss the models found with our approach, and then we compare their performance to other mitigation baselines.

**Setup.** We conducted one NAS+HPO search for each dataset by searching on the train and validation sets. After running these searches, we identified three new candidate architectures for CelebA (SMAC_000, SMAC_010, and SMAC_680), and one candidate for VGGFace2 (SMAC_301) where the naming convention follows that described in Section 4.1. We then retrained each of these models and those high performing models from Section 3 for three seeds to study the robustness of error and disparity for these models; we evaluated their performance on the validation and test sets for each dataset, where we follow the evaluation scheme of Section 3.

**Comparison against `timm` models.** On CelebA (Figure 3), our models Pareto-dominate all of the `timm` models with nontrivial accuracy on the validation set. On the test set, our models still Pareto-dominate all highly competitive models (with Error<0.1), but one of the original configurations (DPN with Magface) also becomes Pareto-optimal. However, the error of this architecture is 0.13, which is significantly higher than our models (0.03-0.04). Also, some models (e.g., VoVNet and DenseNet) show very large standard errors across seeds. Hence, it becomes important to also study

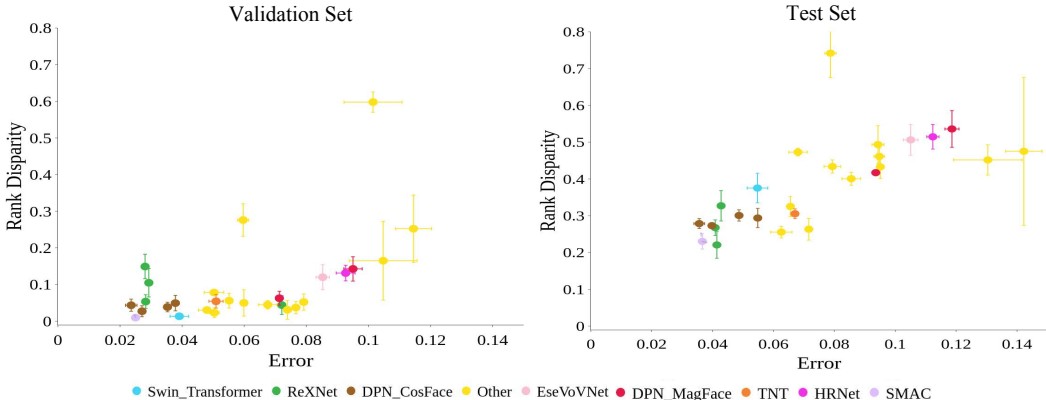

Figure 4: Pareto front of the models discovered by SMAC and the rank-1 models from `timm` for the *(a)* validation and *(b)* test sets on VGGFace2. Each point corresponds to the mean and standard error of an architecture after training for 3 seeds. The SMAC models are Pareto-optimal the top performing `timm` models (Error<0.1).

the robustness of models across seeds along with the accuracy and disparity Pareto front. Finally, on VGGFace2 (Figure 4), our models are also Pareto-optimal for both the validation and test sets.

**Novel Architectures Outperform the State of the Art.** Comparing the results of our automatically-found models to the current state of the art baseline ArcFace [23] in terms of error demonstrates that our strategy clearly establishes a new state of the art. While ArcFace [23] achieves an error of 4.35% with our training pipeline on CelebA, our best-performing novel architecture achieves a much lower error of 3.10%. Similarly, the current VGGFace2 state of the art baseline [112] achieves an error of 4.5%, whereas our best performing novel architecture achieves a much lower error of 3.66%.

**Novel Architectures Pareto-Dominate other Bias Mitigation Strategies.** There are three common pre-, post-, and in-processing bias mitigation strategies in face identification. First, Chang et al. [9] demonstrated that randomly flipping labels in the training data of the subgroup with superior accuracy can yield fairer systems; we call this technique `Flipped`. Next, Wang and Deng [110] use different angular margins during training and therefore promote better feature discrimination for the minority class; we call this technique `Angular`. Finally, Morales et al. [76] introduced `SensitiveNets` which is a sensitive information removal network trained on top of a pre-trained feature extractor with an adversarial sensitive regularizer. While other bias mitigation techniques exist in face recognition, these three are the most used and pertinent to *face identification*. See Cherepanova et al. [14] for an overview of the technical challenges of bias mitigation in face recognition. We take the top performing, Pareto-optimal `timm` models from the previous section and apply the three bias mitigation techniques (`Flipped`, `Angular`, and `SensitiveNets`). We also apply these same techniques to the novel architectures that we found. The results in Table 1 show that the novel architectures from our NAS+HPO-based mitigation strategy Pareto-dominate the bias-mitigated models. In VGGFace2, the SMAC_301 model achieves the best performance, both in terms of error and fairness, compared to the bias-mitigated models. On CelebA, the same is true for the SMAC_680 model.

**NAS+HPO-Based Bias Mitigation can be Combined with other Bias Mitigation Strategies.** Additionally, we combined the three other bias mitigation methods with the SMAC models that resulted from our NAS+HPO-based bias mitigation strategy. More precisely, we first conducted our NAS+HPO approach and then applied the `Flipped`, `Angular`, and `SensitiveNets` approach afterwards. On both datasets, the resulting models continue to Pareto-dominate the other bias mitigation strategies used by themselves and ultimately yield the model with the lowest rank disparity of all the models (0.18 on VGGFace2 and 0.03 on CelebA). In particular, the bias improvement of SMAC_000+`Flipped` model is notable, achieving a score of 0.03 whereas the lowest rank disparity of any model from Figure 3 is 2.63, a 98.9% improvement. In Appendix C.6, we demonstrate that this result is robust to the fairness metric — specifically our bias mitigation strategy Pareto-dominates the other approaches on all five fairness metrics.

Table 1: Comparison of bias mitigation techniques where the SMAC models were found with our NAS+HPO bias mitigation technique and the other three techniques are standard in facial recognition: Flipped [9], Angular [76], and SensitiveNets [110]. Items in bold are Pareto-optimal. The values show (Error;Rank Disparity). Other metrics are reported in Appendix C.6 and Table 8.

| | Trained on VGGFace2 | | | | | Trained on CelebA | | | |
|---|---|---|---|---|---|---|---|---|---|
| Model | Baseline | Flipped | Angular | SensitiveNets | Model | Baseline | Flipped | Angular | SensitiveNets |
| SMAC_301 | **(3.66;0.23)** | **(4.95;0.18)** | (4.14;0.25) | (6.20;0.41) | SMAC_000 | (3.25;2.18) | **(5.20;0.03)** | (3.45;2.28) | (3.45;2.18) |
| DPN | (3.56;0.27) | (5.87;0.32) | (6.06;0.36) | (4.76;0.34) | SMAC_010 | (4.14;2.27) | (12.27; 5.46) | (4.50;2.50) | (3.99;2.12) |
| ReXNet | (4.09;0.27) | (5.73;0.45) | (5.47;0.26) | (4.75;0.25) | SMAC_680 | **(3.22;1.96)** | (12.42;4.50) | (3.80;4.16) | (3.29;2.09) |
| Swin | (5.47;0.38) | (5.75;0.44) | (5.23;0.25) | (5.03;0.30) | ArcFace | (11.30;4.6) | (13.56;2.70) | (9.90;5.60) | (9.10;3.00) |

Table 2: We transfer the evaluation of top performing models on VGGFace2 and CelebA onto six other common face recognition datasets: LFW [53], CFP_FF [100], CFP_FP [100], AgeDB [77], CALFW [128], CPLPW [127]. The novel architectures found with our bias mitigation strategy significantly outperform other models in terms of accuracy. Refer Table 9 for the complete results.

| Architecture (trained on VGGFace2) | LFW | CFP_FF | CFP_FP | AgeDB | CALFW | CPLFW |
|---|---|---|---|---|---|---|
| Rexnet_200 | 82.60 | 80.91 | 65.51 | 59.18 | 68.23 | 62.15 |
| DPN_SGD | 93.0 | 91.81 | 78.96 | 71.87 | 78.27 | 72.97 |
| DPN_AdamW | 78.66 | 77.17 | 64.35 | 61.32 | 64.78 | 60.30 |
| SMAC_301 | **96.63** | **95.10** | **86.63** | **79.97** | **86.07** | **81.43** |
| **Architecture (trained on CelebA)** | **LFW** | **CFP_FF** | **CFP_FP** | **AgeDB** | **CALFW** | **CPLFW** |
| DPN_CosFace | 87.78 | 90.73 | 69.97 | 65.55 | 75.50 | 62.77 |
| DPN_MagFace | 91.13 | 92.16 | 70.58 | 68.17 | 76.98 | 60.80 |
| SMAC_000 | **94.98** | 95.60 | **74.24** | 80.23 | 84.73 | 64.22 |
| SMAC_010 | 94.30 | 94.63 | 73.83 | **80.37** | 84.73 | **65.48** |
| SMAC_680 | 94.16 | **95.68** | 72.67 | 79.88 | **84.78** | 63.96 |

**Novel Architectures Generalize to Other Datasets.** We observed that when transferring our novel architectures to other facial recognition datasets that focus on fairness-related aspects, our architectures consistently outperform other existing architectures by a significant margin. We take the state-of-the-art models from our experiments and test the weights from training on CelebA and VGGFace2 on different datasets which the models did not see during training. Specifically, we transfer the evaluation of the trained model weights from CelebA and VGGFace2 onto the following datasets: LFW [53], CFP_FF [100], CFP_FP [100], AgeDB [77], CALFW [128], CPLFW [127]. Table 2 demonstrates that our approach consistently achieves the highest performance among various architectures when transferred to other datasets. This finding indicates that our approach exhibits exceptional generalizability compared to state-of-the-art face recognition models in terms of transfer learning to diverse datasets.

**Novel Architectures Generalize to Other Sensitive Attributes.** The superiority of our novel architectures even goes beyond accuracy-related metrics when transferring to other datasets — our novel architectures have superior fairness properties compared to the existing architectures *even on datasets which have completely different protected attributes than were used in the architecture search*. Specifically, to inspect the generalizability of our approach to other protected attributes, we transferred our models pre-trained on CelebA and VGGFace2 (which have a gender presentation category) to the RFW dataset [111] which includes a protected attribute for race and the AgeDB dataset [77] which includes a protected attribute for age. The results detailed in Appendix C.7 show that our novel architectures always outperforms the existing architectures, across all five fairness metrics studied in this work on both datasets.

**Novel Architectures Have Less Linear-Separability of Protected Attributes.** Our comprehensive evaluation of multiple face recognition benchmarks establishes the importance of architectures for fairness in face-recognition. However, it is natural to wonder: *"What makes the discovered architectures fair in the first place?"* To answer this question, we use linear probing to dissect the

Table 3: Linear Probes on architectures. Lower gender classification accuracy is better

| Architecture (trained on VGGFace2) | Accuracy on Layer N ↓ | Accuracy on Layer N-1 ↓ |
|---|---|---|
| DPN_MagFace_SGD | 86.042% | 95.461% |
| DPN_CosFace_SGD | 90.719% | 93.787% |
| DPN_CosFace_AdamW | 87.385% | 94.444% |
| SMAC_301 | **69.980**% | **68.240**% |

intermediate features of our searched architectures and DPNs, which our search space is based upon. Intuitively, given that our networks are trained only on the task of face recognititon, we do not want the intermediate feature representations to implicitly exploit knowledge about protected attributes (e.g., gender). To this end we insert linear probes[1] at the last two layers of different Pareto-optimal DPNs and the model obtained by our NAS+HPO-based bias mitigation. Specifically, we train an MLP on the feature representations extracted from the pre-trained models and the protected attributes as labels and compute the gender-classification accuracy on a held-out set. We consider only the last two layers, so k assumes the values of $N$ and $N-1$ with $N$ being the number of layers in DPNs (and the searched models). We represent the classification probabilities for the genders by $gp_k = softmax(W_k + b)$, where $W_k$ is the weight matrix of the $k$-th layer and $b$ is a bias. We provide the classification accuracies for the different pre-trained models on VGGFace2 in Table 3. This demonstrates that, as desired, our searched architectures maintain a lower classification accuracy for the protected attribute. In line with this observation, in the t-SNE plots in Figure 18 in the appendix, the DPN displays a higher degree of separability of features.

**Comparison between different NAS+HPO techniques** We also perform an ablation across different multi-objective NAS+HPO techniques. Specifically we compare the architecture derived by SMAC with architectures derived by the evolutionary multi-objective optimization algorithm NSGA-II [22] and multi-objective asynchronous successive halving (MO-ASHA) [98]. We obesrve that the architecture derived by SMAC Pareto-dominates the other NAS methods in terms of accuracy and diverse fairness metrics Table 4. We use the implementation of NSGA-II and MO-ASHA from the syne-tune library [95] to perform an ablation across different baselines.

Table 4: Comparison between architectures derived by SMAC and other NAS baselines

| NAS Method | Accuracy ↑ | Rank Disparity↓ | Disparity↓ | Ratio↓ | Rank Ratio ↓ | Error Ratio↓ |
|---|---|---|---|---|---|---|
| MO-ASHA_108 | 95.212 | 0.408 | 0.038 | 0.041 | 0.470 | 0.572 |
| NSGA-II_728 | 86.811 | 0.599 | 0.086 | 0.104 | 0.490 | **0.491** |
| SMAC_301 | **96.337** | **0.230** | **0.030** | **0.032** | **0.367** | 0.582 |

# 5 Conclusion, Future Work and Limitations

**Conclusion.** Our approach studies a novel direction for bias mitigation by altering network topology instead of loss functions or model parameters. We conduct the first large-scale analysis of the relationship among hyperparameters and architectural properties, and accuracy, bias, and disparity in predictions across large-scale datasets like CelebA and VGGFace2. Our bias mitigation technique centering around Neural Architecture Search and Hyperparameter Optimization is very competitive compared to other common bias mitigation techniques in facial recognition.

Our findings present a paradigm shift by challenging conventional practices and suggesting that seeking a fairer architecture through search is more advantageous than attempting to rectify an unfair one through adjustments. The architectures obtained by our joint NAS and HPO generalize across different face recognition benchmarks, different protected attributes, and exhibit lower linear-separability of protected attributes.

**Future Work.** Since our work lays the foundation for studying NAS+HPO for fairness, it opens up a plethora of opportunities for future work. We expect the future work in this direction to focus on

studying different multi-objective algorithms [34, 60] and NAS techniques [67, 125, 115] to search for inherently fairer models. Further, it would be interesting to study how the properties of the architectures discovered translate across different demographics and populations. Another potential avenue for future work is incorporating priors and beliefs about fairness in the society from experts to further improve and aid NAS+HPO methods for fairness. Given the societal importance, it would be interesting to study how our findings translate to real-life face recognition systems under deployment. Finally, it would also be interesting to study the degree to which NAS+HPO can serve as a general bias mitigation strategy beyond the case of facial recognition.

**Limitations.** While our work is a step forward in both studying the relationship among architectures, hyperparameters, and bias, and in using NAS techniques to mitigate bias in face recognition models, there are important limitations to keep in mind. Since we only studied a few datasets, our results may not generalize to other datasets and fairness metrics. Second, since face recognition applications span government surveillance [49], target identification from drones [72], and identification in personal photo repositories [36], our findings need to be studied thoroughly across different demographics before they could be deployed in real-life face recognition systems. Furthermore, it is important to consider how the mathematical notions of fairness used in research translate to those actually impacted [94], which is a broad concept without a concise definition. Before deploying a particular system that is meant to improve fairness in a real-life application, we should always critically ask ourselves whether doing so would indeed prove beneficial to those impacted by the given sociotechnical system under consideration or whether it falls into one of the traps described by Selbst et al. [99]. Additionally, work in bias mitigation, writ-large and including our work, can be certainly encourage techno-solutionism which views the reduction of statistical bias from algorithms as a justification for their deployment, use, and proliferation. This of course can have benefits, but being able to reduce the bias in a technical system is a *different question* from whether a technical solution *should* be used on a given problem. We caution that our work should not be interpreted through a normative lens on the appropriateness of using facial recognition technology.

In contrast to some other works, we do, however, feel, that our work helps to overcome the portability trap [99] since it empowers domain experts to optimize for the right fairness metric, in connection with public policy experts, for the problem at hand rather than only narrowly optimizing one specific metric. Additionally, the bias mitigation strategy which we propose here can be used in other domains and applied to applications which have more widespread and socially acceptable algorithmic applications [19].

### Acknowledgments

This research was partially supported by the following sources: NSF CAREER Award IIS-1846237, NSF D-ISN Award #2039862, NSF Award CCF-1852352, NIH R01 Award NLM-013039-01, NIST MSE Award #20126334, DARPA GARD #HR00112020007, DoD WHS Award #HQ003420F0035, ARPA-E Award #4334192; TAILOR, a project funded by EU Horizon 2020 research and innovation programme under GA No 952215; the German Federal Ministry of Education and Research (BMBF, grant RenormalizedFlows 01IS19077C); the Deutsche Forschungsgemeinschaft (DFG, German Research Foundation) under grant number 417962828; the European Research Council (ERC) Consolidator Grant "Deep Learning 2.0" (grant no. 101045765). Funded by the European Union. Views and opinions expressed are however those of the author(s) only and do not necessarily reflect those of the European Union or the ERC. Neither the European Union nor the ERC can be held responsible for them.

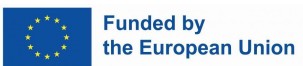

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

# A  Ethics Statement

Face recognition systems are being used for more and more parts of daily lives, from government surveillance [49], to target identification from drones [72], to identification in personal photo repositories [36]. It is also increasingly evident that many of these models are biased based on race and gender [37, 92, 91]. If left unchecked, these technologies, which make biased decision for life-changing events, will only deepen existing societal harms. Our work seeks to better understand and mitigate the negative effects that biased face recognition models have on society. By conducting the first large-scale study of the effect of architectures and hyperparameters on bias, and by developing and open-sourcing face recognition models that are more fair than all other competitive models, we provide a resource for practitioners to understand inequalities inherent in face recognition systems and ultimately advance fundamental understandings of the harms and technological ills of these systems.

That said, we would like to address potential ethical challenges of our work. We believe that the main ethical challenge of this work centers on our use of certain datasets. We acknowledge that the common academic datasets which we used to evaluate our research questions — CelebA [69] and VGGFace2 [8] — are all datasets of images scraped from the web without the informed consent of those whom are depicted. This also includes the datasets we transfer to, including LFW, CFP_FF, CFP_FP, AgeDB, CALFW, CPLFW, and RFW. This ethical challenge is one that has plagued the research and computer vision community for the last decade [86, 85] and we are excited to see datasets being released which have fully informed consent of the subjects, such as the Casual Conversations Dataset [45]. Unfortunately, this dataset in particular has a rather restrictive license, much more restrictive than similar datasets, which prohibited its use in our study. Additionally, these datasets all exhibit representational bias where the categories of people that are included in the datasets are not equally represented. This can cause many problems; see [14] for a detailed look at reprsentational bias's impact in facial recognition specifically. At least during training, we did address representaional bias by balancing the training data between gender presentations appropriately.

We also acknowledge that while our study is intended to be constructive in performing the first neural architecture search experiments with fairness considerations, the specific ethical challenge we highlight is that of unequal or unfair treatment by the technologies. We note that our work could be taken as a litmus test which could lead to the further proliferation of facial recognition technology which could cause other harms. If a system demonstrates that it is less biased than other systems, this could be used as a reason for the further deployment of facial technologies and could further impinge upon unwitting individual's freedoms and perpetuate other technological harms. We explicitly caution against this form of techno-solutionism and do not want our work to contribute to a conversation about whether facial recognition technologies *should* be used by individuals.

Experiments were conducted using a private infrastructure, which has a carbon efficiency of 0.373 kgCO$_2$eq/kWh. A cumulative of 88 493 hours of computation was performed on hardware of type RTX 2080 Ti (TDP of 250W). Total emissions are estimated to be 8,251.97 kgCO$_2$eq of which 0% was directly offset. Estimations were conducted using the MachineLearning Impact calculator presented in [58]. By releasing all of our raw results, code, and models, we hope that our results will be widely beneficial to researchers and practitioners with respect to designing fair face recognition systems.

# B  Reproducibility Statement

We ensure that all of our experiments are reproducible by releasing our code and raw data files at `https://github.com/dooleys/FR-NAS`. We also release the instructions to reproduce our results with the code. Furthermore, we release all of the configuration files for all of the models trained. Our experimental setup is described in Section 3 and Appendix C.1. We provide clear documentation on the installation and system requirements in order to reproduce our work. This includes information about the computing environment, package requirements, dataset download procedures, and license information. We have independently verified that the experimental framework is reproducible which should make our work and results and experiments easily accessible to future researchers and the community.

Table 5: The fairness metrics explored in this paper. Rank Disparity is explored in the main paper and the other metrics are reported in Appendix C.4

| Fairness Metric | Equation |
| --- | --- |
| Rank Disparity | $|\text{Rank}(male) - \text{Rank}(female)|$ |
| Disparity | $|\text{Accuracy}(male) - \text{Accuracy}(female)|$ |
| Ratio | $|1 - \frac{\text{Accuracy}(male)}{\text{Accuracy}(female)}|$ |
| Rank Ratio | $|1 - \frac{\text{Rank}(male)}{\text{Rank}(female)}|$ |
| Error Ratio | $|1 - \frac{\text{Error}(male)}{\text{Error}(female)}|$ |

## C   Further Details on Experimental Design and Results

### C.1   Experimental Setup

The list of the models we study from `timm` are: `coat_lite_small` [120], `convit_base` [20], `cspdarknet53` [107], `dla102x2` [122], `dpn107` [10], `ese_vovnet39b` [62], `fbnetv3_g` [18], `ghostnet_100` [42], `gluon_inception_v3` [103], `gluon_xception65` [15], `hrnet_w64` [102], `ig_resnext101_32x8d` [119], `inception_resnet_v2` [104], `inception_v4` [104], `jx_nest_base` [126], `legacy_senet154` [51], `mobilenetv3_large_100` [50], `resnetrs101` [4], `rexnet_200` [41], `selecsls60b` [74], `swin_base_patch4_window7_224` [68], `tf_efficientnet_b7_ns`' [105], '`tnt_s_patch16_224`[43], `twins_svt_large` [16] , `vgg19` [101], `vgg19_bn` [101], `visformer_small` [11], `xception` and `xception65` [15].

We study at most 13 configurations per model ie 1 default configuration corresponding to the original model hyperparameters with CosFace as head. Further, we have at most 12 configs consisting of the 3 heads (CosFace, ArcFace, MagFace) $\times$ 2 learning rates(0.1,0.001) $\times$ 2 optimizers (SGD, AdamW). All the other hyperparameters are held constant for training all the models. All model configurations are trained with a total batch size of 64 on 8 RTX2080 GPUS for 100 epochs each.

We study these models across five important fairness metrics in face identification: Rank Disparity, Disparity, Ratio, Rank Ratio, and Error Ratio. Each of these metrics is defined in Table 5.

### C.2   Additional details on NAS+HPO Search Space

We replace the repeating `1x1_conv-3x3_conv-1x1_conv` block in Dual Path Networks with a simple recurring searchable block. depicted in Figure 6. Furthermore, we stack multiple such searched blocks to closely follow the architecture of Dual Path Networks. We have nine possible choices for each of the three operations in the DPN block, each of which we give a number 0 through 8, depicted in Figure 6. The choices include a vanilla convolution, a convolution with pre-normalization and a convolution with post-normalization Table 7. To ensure that all the architectures are tractable in terms of memory consumption during search we keep the final projection layer (to 1000 dimensionality) in `timm`.

### C.3   Obtained architectures and hyperparameter configurations from Black-Box-Optimization

In Figure 5 we present the architectures and hyperparameters discovered by SMAC. Particularly we observe that `conv 3x3` followed `batch norm` is a preferred operation and CosFace is the preferred head/loss choice.

### C.4   Analysis of the Pareto front of different Fairness Metrics

In this section, we include additional plots that support and expand on the main paper. Primarily, we provide further context of the Figures in the main body in two ways. First, we provide replication plots of the figures in the main body but for all models. Recall, the plots in the main body only show models with Error<0.3, since high performing models are the most of interest to the community.

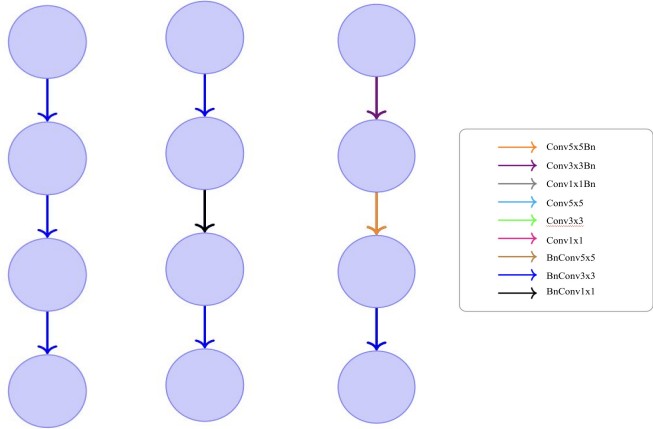

Figure 5: SMAC discovers the above building blocks with (a) corresponding to architecture with CosFace, with SGD optimizer and learning rate of 0.2813 as hyperparamters (b) corresponding to CosFace, with SGD as optimizer and learning rate of 0.32348 and (c) corresponding to CosFace, with AdamW as optimizer and learning rate of 0.0006

Table 6: Searchable hyperparameter choices.

| Hyperparameter | Choices |
| --- | --- |
| Architecture Head/Loss | MagFace, ArcFace, CosFace |
| Optimizer Type | Adam, AdamW, SGD |
| Learning rate (conditional) | Adam/AdamW $\rightarrow [1e-4, 1e-2]$, SGD $\rightarrow [0.09, 0.8]$ |

Table 7: Operation choices and definitions.

| Operation Index | Operation | Definition |
| --- | --- | --- |
| 0 | BnConv1x1 | Batch Normalization $\rightarrow$ Convolution with 1x1 kernel |
| 1 | Conv1x1Bn | Convolution with 1x1 kernel $\rightarrow$ Batch Normalization |
| 2 | Conv1x1 | Convolution with 1x1 kernel |
| 3 | BnConv3x3 | Batch Normalization $\rightarrow$ Convolution with 3x3 kernel |
| 4 | Conv3x3Bn | Convolution with 3x3 kernel $\rightarrow$ Batch Normalization |
| 5 | Conv3x3 | Convolution with 3x3 kernel |
| 6 | BnConv5x5 | Batch Normalization $\rightarrow$ Convolution with 5x5 kernel |
| 7 | Conv5x5Bn | Convolution with 5x5 kernel $\rightarrow$ Batch Normalization |
| 8 | Conv5x5 | Convolution with 5x5 kernel |

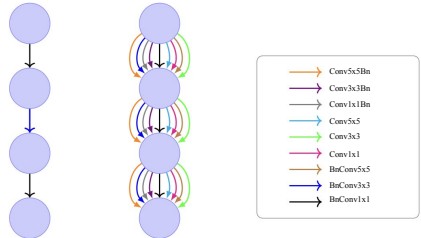

Figure 6: DPN block (left) vs. our searchable block (right).

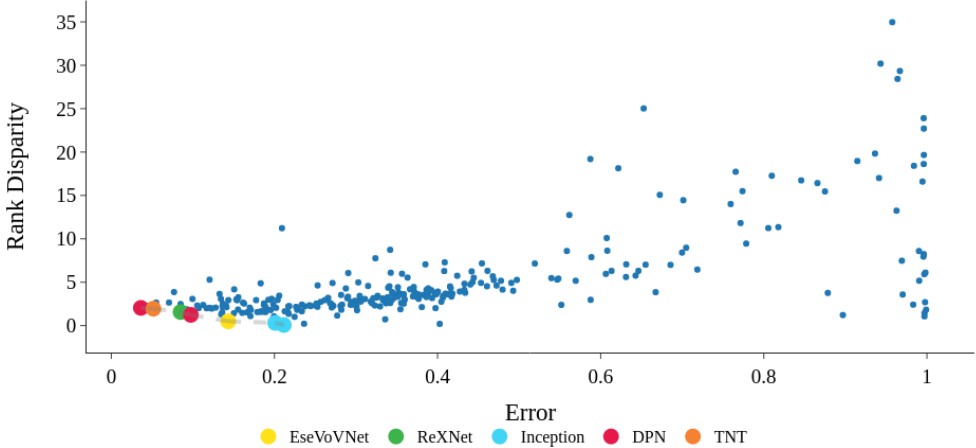

Figure 7: Replication of CelebA Figure 2 with all data points. Error-Rank Disparity Pareto front of the architectures with any non-trivial error. Models in the lower left corner are better. The Pareto front is notated with a dashed line. Other points are architecture and hyperparameter combinations which are not Pareto-dominant.

VGGFace2 Validation Set

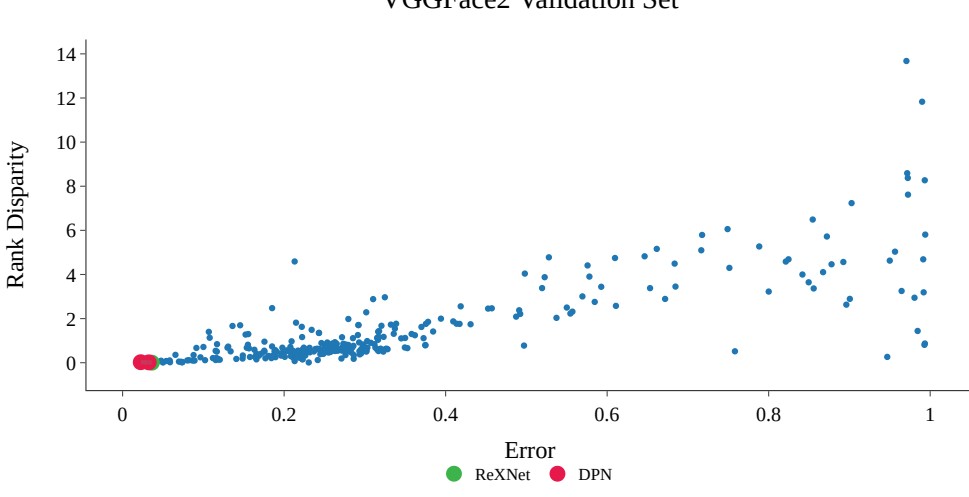

Figure 8: Replication of VGGFace2 Figure 2 with all data points. Error-Rank Disparity Pareto front of the architectures with any non-trivial error. Models in the lower left corner are better. The Pareto front is notated with a dashed line. Other points are architecture and hyperparameter combinations which are not Pareto-dominant.

Second, we also show figures which depict other fairness metrics used in facial identification. The formulas for these additional fairness metrics can be found in Table 5.

We replicate Figure 2 in Figure 7 and Figure 8. We add additional metrics for CelebA in Figure 9-Figure 11 and for VGGFace2 in Figure 12-Figure 15.

### C.5 Correlation between fairness and model statistics

We examine the correlation of each fairness-metric with model statistics. We compute statistics like number of parameters, model latency, number of convolutions, number of linear layers, and number of batch-norms in a model's definition. Interestingly, we observe very low and non-significant

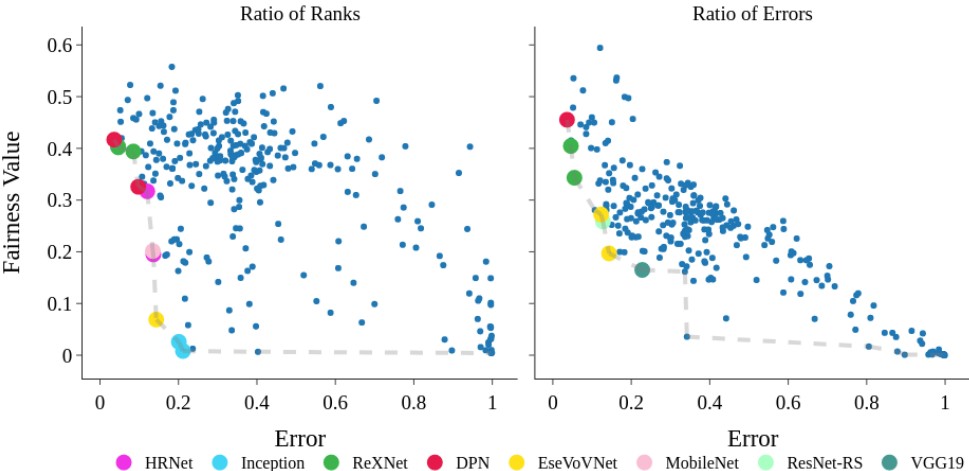

Figure 9: Replication of Figure 7 on the CelebA validation dataset with Ratio of Ranks (left) and Ratio of Errors (right) metrics.

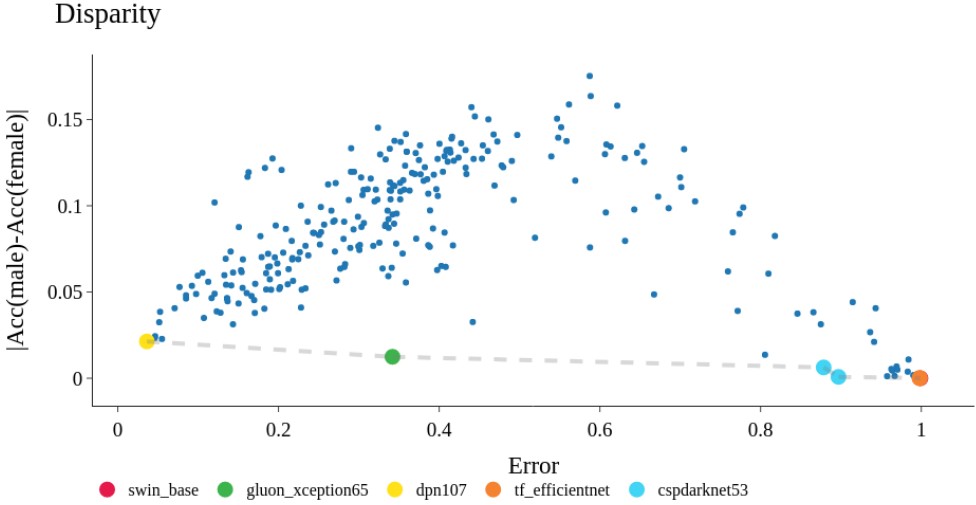

Figure 10: Replication of Figure 7 on the CelebA validation dataset with the Disparity in accuracy metric.

correlations between parameter sizes and different fairness metrics Figure 16. This observation supports the claim that increases in accuracy and decreases in disparity are very closely tied to the architectures and feature representations of the model, irrespective of the parameter size of the model. Hence, we do not constraint model parameter sizes to help our NAS+HPO approach search in a much richer search space.

## C.6   Comparison to other Bias Mitigation Techniques on all Fairness Metrics

We have shown that our bias mitigation approach Pareto-dominates the existing bias mitigation techniques in face identification on the Rank Disparity metric. Here, we perform the same experiments but evaluate on the four other metrics discussed in the face identification literature: Disparity, Rank Ratio, Ratio, and Error Ratio.

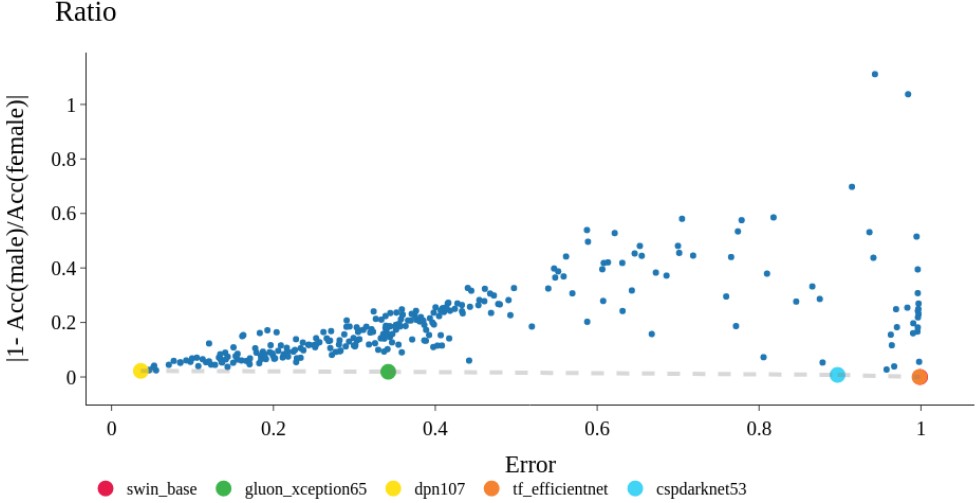

Figure 11: Replication of Figure 7 on the CelebA validation dataset with the Ratio in accuracy metric.

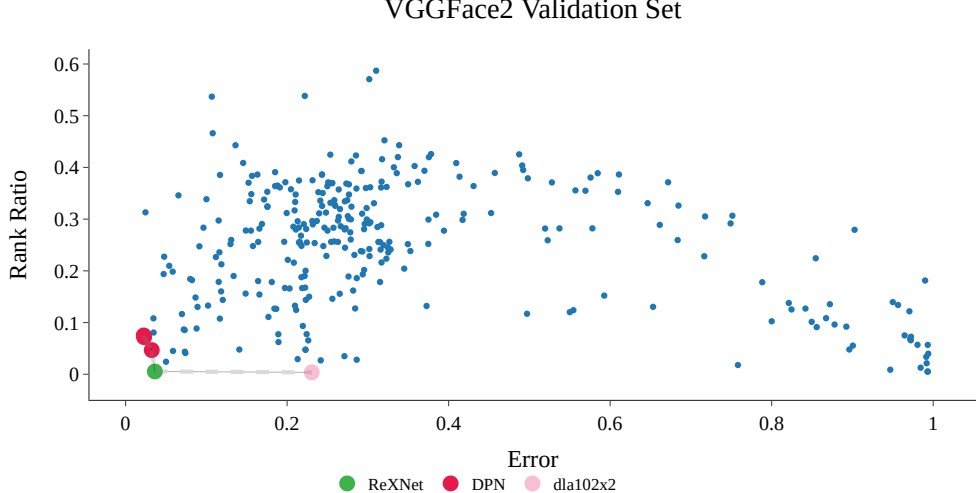

Figure 12: Replication of Figure 8 on the VGGFace2 validation dataset with Ratio of Ranks metric.

Recall, we take top performing, Pareto-optimal models from Section 4 and apply the three bias mitigation techniques: Flipped, Angular, and SensitiveNets. We also apply these same techniques to the novel architectures that we found. We report results in Table 1.

In Table 8, we see that in every metric, the SMAC_301 architecture is Pareto-dominant and that the SMAC_301, demonstrating the robustness of our approach.

## C.7   Transferability to other Sensitive Attributes

The superiority of our novel architectures even goes beyond accuracy when transfering to other datasets — our novel architectures have superior fairness property compared to the existing architectures **even on datasets which have completely different protected attributes than were used in the architecture search**. To inspect the generalizability of our approach to other protected attributes, we transferred our models pre-trained on CelebA and VGGFace2 (which have a gender presentation category) to the RFW dataset [111] which includes a protected attribute for race. We see that our novel architectures always outperforms the existing architectures across all five fairness metrics studied in this work. See Table 10 for more details on each metric. We see example Pareto fronts for

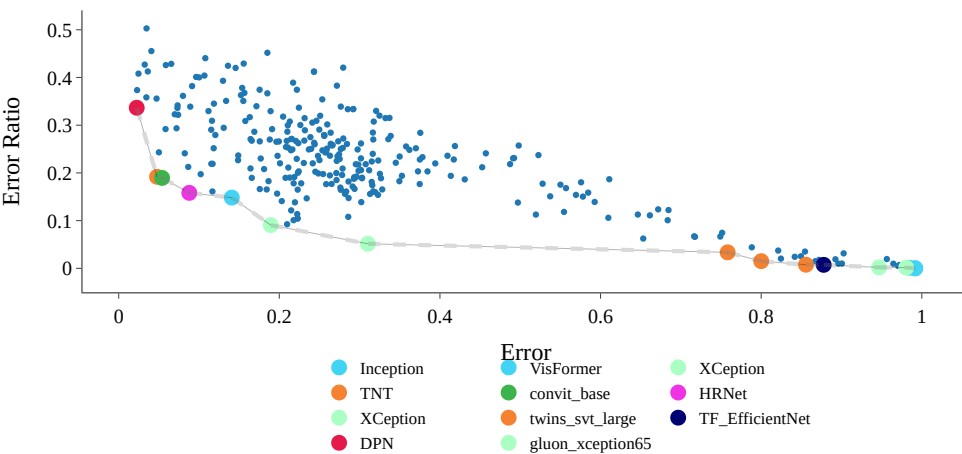

Figure 13: Replication of Figure 8 on the VGGFace2 validation dataset with Ratio of Errors metric.

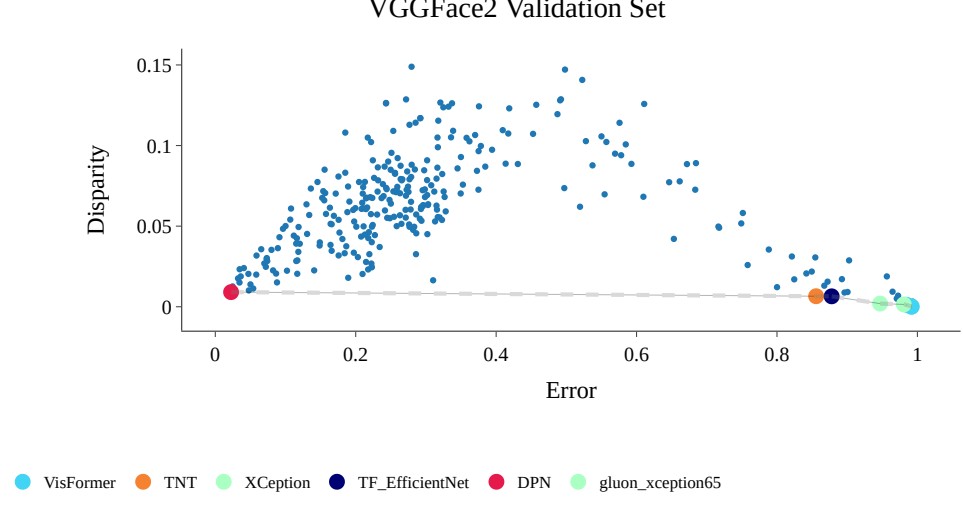

Figure 14: Replication of Figure 8 on the VGGFace2 validation dataset with the Disparity in accuracy metric.

these transfers in Figure 17. They are always on the Pareto front for all fairness metrics considered, and mostly Pareto-dominate all other architectures on this task. In this setting, since the race label in RFW is not binary, the Rank Disparity metric considered in Table 10 and Figure 17 is computed as the maximum rank disparity between pairs of race labels.

Furthermore, we also evaluate the transfer of fair properties of our models across different age groups on the AgeDB dataset [77]. In this case we use age as a protected attribute. We group the faces into 4 age groups 1-25, 26-50, 51-75 and 76-110. Then we compute the max disparity across age groups. As observed in Table 11 the models discovered by NAS and HPO, Pareto-dominate other competitive hand-crafted models. This further emphasise the generalizabitlity of the fair features learnt by these models.

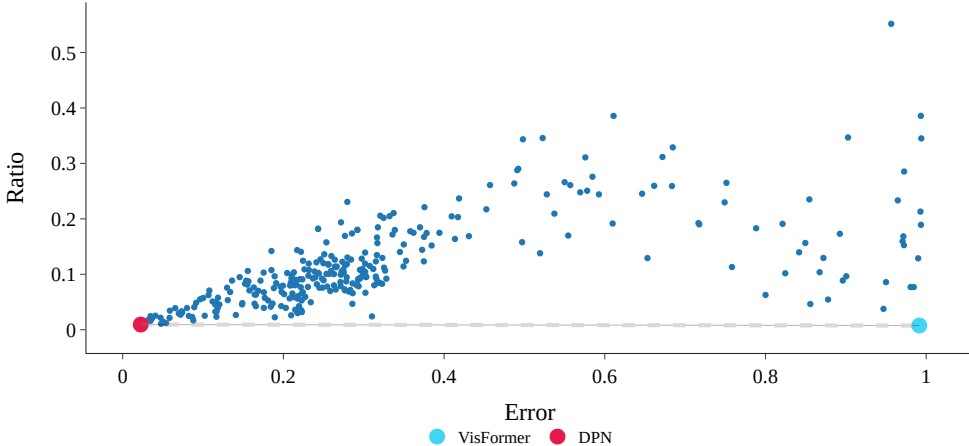

Figure 15: Replication of Figure 8 on the VGGFace2 validation dataset with the Ratio in accuracy metric.

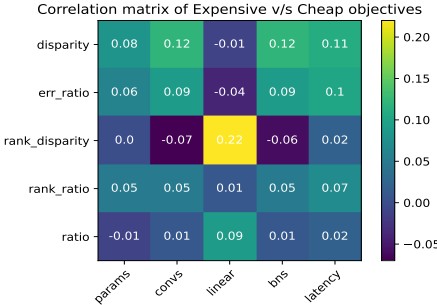

Figure 16: Correlation map between different fairness metrics and architecture statistics. We find no significant correlation between these objectives, e.g. between fairness metrics and parameter count.

Table 11: Taking the highest performing models from the Pareto front of both VGGFace2 and CelebA, we transfer their evaluation onto a dataset with a different protected attribute – age – on the AgeDB dataset [77]. The novel architectures which we found with our bias mitigation strategy are Pareto-dominant with respect to the Accuracy and Disparity metrics

| Architecture (trained on VGGFace2) | Overall Accuracy ↑ | Disparity ↓ |
|---|---|---|
| Rexnet_200 | 59.18 | 28.9150 |
| DPN_SGD | 71.87 | 22.4633 |
| DPN_AdamW | 61.32 | 21.1437 |
| SMAC_301 | **79.97** | **18.827** |
| **Architecture (trained on CelebA)** | **Accuracy** | **Disparity** |
| DPN_CosFace | 65.55 | 27.2434 |
| DPN_MagFace | 68.17 | 31.2903 |
| SMAC_000 | 80.23 | **19.6481** |
| SMAC_010 | **80.37** | 26.5103 |
| SMAC_680 | 79.88 | 20.0586 |

Table 8: Comparison bias mitigation techniques where the SMAC models were found on VGGFace2 with NAS bias mitigation technique and the other three techniques are standard in facial recognition: Flipped [9], Angular [76], and Discriminator [110]. Items in bold are Pareto-optimal. The values show (Error;*metric*).

| Model | Rank Disparity | | | | Disparity | | | |
|---|---|---|---|---|---|---|---|---|
| | Baseline | Flipped | Angular | SensitiveNets | Baseline | Flipped | Angular | SensitiveNets |
| SMAC_301 | **(3.66;0.23)** | **(4.95;0.18)** | (4.14;0.25) | (6.20;0.41) | **(3.66;0.03)** | **(4.95;0.02)** | (4.14;0.04) | (6.14;0.04) |
| DPN | (3.56;0.27) | (5.87;0.32) | (6.06;0.36) | (4.76;0.34) | (3.98;0.04) | (5.87;0.05) | (6.06;0.05) | (4.78;0.05) |
| ReXNet | (4.09;0.27) | (5.73;0.45) | (5.47;0.26) | (4.75;0.25) | (4.09;0.03) | (5.73;0.05) | (5.47;0.05) | (4.75;0.04) |
| Swin | (5.47;0.38) | (5.75;0.44) | (5.23;0.25) | (5.03;0.30) | (5.47;0.05) | (5.75;0.04) | (5.23;0.04) | (5.03;0.04) |

| Model | Rank Ratio | | | | Ratio | | | |
|---|---|---|---|---|---|---|---|---|
| | Baseline | Flipped | Angular | SensitiveNets | Baseline | Flipped | Angular | SensitiveNets |
| SMAC_301 | **(3.66;0.37)** | **(4.95;0.21)** | (4.14;0.39) | (6.14;0.41) | **(3.66;0.03)** | **(4.95;0.02)** | (4.14;0.04) | (6.14;0.05) |
| DPN | (3.98;0.49) | (5.87;0.49) | (6.06;0.54) | (4.78;0.49) | (3.98;0.04) | (5.87;0.06) | (6.06;0.06) | (4.78;0.05) |
| ReXNet | (4.09;0.41) | (5.73;0.53) | (5.47;0.38) | (4.75;0.34) | (4.09;0.04) | (5.73;0.05) | (5.47;0.05) | (4.75;0.04) |
| Swin | (5.47;0.47) | (5.75;0.47) | (5.23;0.42) | (5.03;0.43) | (5.47;0.05) | (5.75;0.05) | (5.23;0.05) | (5.03;0.05) |

| Model | Error Ratio | | | |
|---|---|---|---|---|
| | Baseline | Flipped | Angular | SensitiveNets |
| SMAC_301 | **(3.66;0.58)** | **(4.95;0.29)** | (4.14;0.60) | (6.14;0.52) |
| DPN | (3.98;0.65) | (5.87;0.62) | (6.06;0.62) | (4.78;0.69) |
| ReXNet | (4.09;0.60) | (5.73;0.57) | (5.47;0.59) | (4.75;0.58) |
| Swin | (5.47;0.60) | (5.75;0.56) | (5.23;0.60) | (5.03;0.60) |

Table 9: Taking the highest performing models from the Pareto front of both VGGFace2 and CelebA, we transfer their evaluation onto six other common face recognition datasets: LFW [53], CFP_FF [100], CFP_FP [100], AgeDB [77], CALFW [128], CPLPW [127]. The novel architectures which we found with our bias mitigation strategy significantly out perform all other models.

| Architecture (trained on VGGFace2) | LFW | CFP_FF | CFP_FP | AgeDB | CALFW | CPLPW |
|---|---|---|---|---|---|---|
| Rexnet_100 | 82.60 | 80.91 | 65.51 | 59.18 | 68.23 | 62.15 |
| DPN_SGD | 93.00 | 91.81 | 78.96 | 71.87 | 78.27 | 72.97 |
| DPN_AdamW | 78.66 | 77.17 | 64.35 | 61.32 | 64.78 | 60.30 |
| SMAC_301 | **96.63** | **95.10** | **86.63** | **79.97** | **86.07** | **81.43** |

| Architecture (trained on CelebA) | LFW | CFP_FF | CFP_FP | AgeDB | CALFW | CPLFW |
|---|---|---|---|---|---|---|
| Rexnet_200 | 71.533 | 73.528 | 54.30 | 55.933 | 60.517 | 52.25 |
| DPN_CosFace | 87.78 | 90.73 | 69.97 | 65.55 | 75.50 | 62.77 |
| DPN_MagFace | 91.13 | 92.16 | 70.58 | 68.17 | 76.98 | 60.80 |
| DenseNet161 | 82.60 | 80.16 | 64.07 | 56.93 | 66.36 | 59.35 |
| Ese_Vovnet39 | 77.00 | 76.50 | 64.68 | 49.38 | 61.283 | 61.016 |
| SMAC_000 | **94.98** | 95.60 | **74.24** | 80.23 | 84.73 | 64.22 |
| SMAC_010 | 94.30 | 94.63 | 73.83 | **80.37** | 84.73 | **65.48** |
| SMAC_680 | 94.16 | **95.68** | 72.67 | 79.88 | **84.78** | 63.96 |

Table 10: Taking the highest performing models from the Pareto front of both VGGFace2 and CelebA, we transfer their evaluation onto a dataset with a different protected attribue – race – on the RFW dataset [111]. The novel architectures which we found with our bias mitigation strategy are always on the Pareto front, and mostly Pareto-dominant of the traditional architectures.

| Fairness Metric | Transfer from CelebA | Transfer from VGGFace2 |
|---|---|---|
| Rank Disparity | Pareto-dominant | Pareto-optimal |
| Disparity | Pareto-dominant | Pareto-dominant |
| Rank Ratio | Pareto-optimal | Pareto-optimal |
| Ratio | Pareto-dominant | Pareto-dominant |
| Error Ratio | Pareto-optimal | Pareto-optimal |

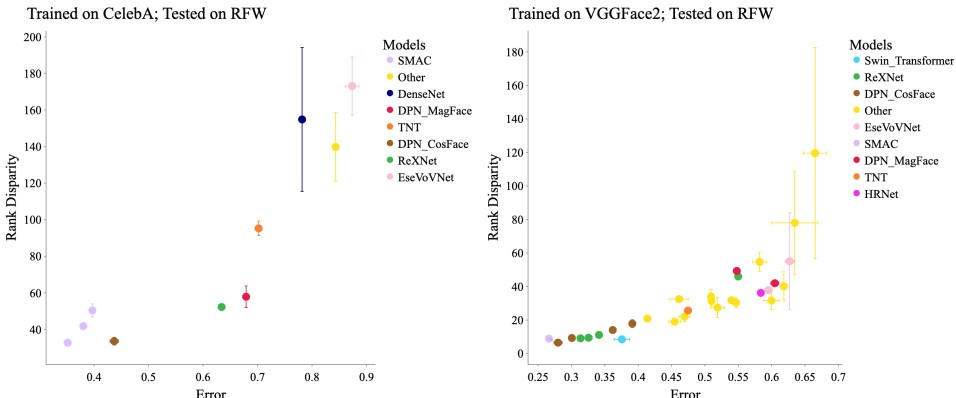

Figure 17: Models trained on CelebA (left) and VGGFace2 (right) evaluated on a dataset with a different protected attribute, specifically on RFW with the racial attribute, and with the Rank Disparity metric. The novel architectures out perform the existing architectures in both settings.

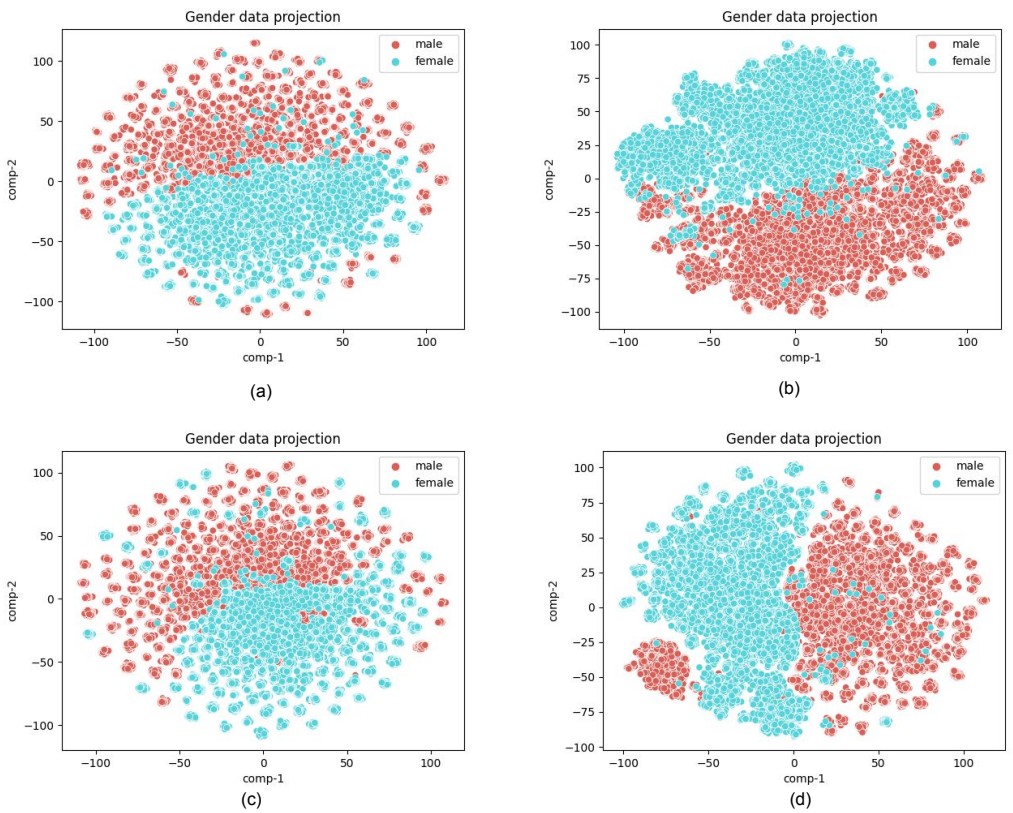

Figure 18: TSNE plots for models pretrained on VGGFace2 on the test-set *(a)* SMAC model last layer *(b)* DPN MagFace on the last layer *(b)* SMAC model second last layer *(b)* DPN MagFace on the second last layer. Note the better linear separability for DPN MagFace in comparison with the SMAC model

