# OpenReview forum: "Rethinking Bias Mitigation: Fairer Architectures Make for Fairer Face Recognition"
_NeurIPS.cc/2023/Conference — NeurIPS 2023 oral_

### Official Review · Reviewer_3GhH · 2023-07-04

**Soundness:** 4 excellent
**Presentation:** 4 excellent
**Contribution:** 4 excellent
**Rating:** 8
**Confidence:** 4

**Summary:**

The paper aims at a fairer face recognition model.

First, the authors conduct large-scale experiments to show that architectures and hyperparameters matter for fairness (Section 3). Concretely, a wide range of models in different architectures and hyperparameters are evaluated in terms of performance (metric: “Error”) and fairness (metric: Rank Disparity), showing that some models (e.g., DPN) are indeed Pareto-optimal compared to others.

Motivated by this finding, unlike previous bias mitigation strategies based on a fixed neural architecture and a set of hyperparameters, the paper provides a new angle on bias mitigation by searching for fairer neural architectures and hyperparameters.

The paper designs a search strategy to satisfy three desiderata: (1) both architectures and hyperparameters are optimized, (2) both accuracy and fairness are used as the objective, and (3) the searching process should be efficient. To this end, the paper uses some existing approaches, such as SMAC, Hyperband, and ParEGO.

The results show that the proposed method is Pareto-optimal compared to existing methods on two face datasets. Furthermore, the experiments show that the proposed method can also generalize to other datasets and protected attributes.

**Strengths:**

1. The proposed method is well-motivated by the experiments in Section 3 to show that architectures and hyperparameters matter for fairness.
2. The paper gives a novel angle from architectures and hyperparameters toward bias mitigation.
3. The paper provides insights into why this method works from the perspective of linear separability of protected attributes (L293).
4. The experiments are extensive.
5. In terms of the results, the proposed method is Pareto-optimal, beating existing bias mitigation methods.
6. The code is provided for better reproducibility.
7. The paper is well-written and easy to follow.


**Weaknesses:**

## Generalization of Pareto-optimal results to different datasets
I appreciate the results of cross-dataset generalization. However, Table 2 only shows the performance results. Therefore, whether or not the proposed method is still Pareto-optimal remains unknown.


### Minor Comments:

The plots in Figure 2-4 are in low resolution. I suggest the authors export the plots in PDF, SVG, or EPS formats instead of image formats (e.g., jpeg).

Fonts in Table 1 are in a strange aspect ratio. I suggest the authors use the “adjustbox” package to adjust the table size.

L300: probes[1] -> probes [1]

Appendix, L806, L808: broken \ref link to the figure. “Figure ??” -> Figure 16

Appendix, Caption of Figure 17: “(b) SMAC model second last layer (b) DPN MagFace on the second last layer” -> “(c) SMAC model second last layer (d) DPN MagFace on the second last layer.”

**Questions:**

1. Can the authors add the fairness-performance results of cross-dataset generalization instead of only showing the performance results?
2. I appreciate the authors’ efforts in explaining why the proposed method work (L293). However, in terms of neural architecture, is there any pattern that makes some architectures more Pareto-optimal than others? This would be interesting because future works may use such a pattern to manually design a fairer architecture.


**Limitations:**

 The authors have adequately addressed the limitations (L333-348). From my perspective, the paper does not have a potential negative societal impact.

---

> ### Author Rebuttal · Authors · 2023-08-09
>
> Thank you for your time and thoughtful feedback on our manuscript. We appreciate that you find our approach well-motivated, our angle of architectures and hyperparameters novel, our experiments extensive, our method reproducible and the paper well-written and easy to follow. We address each of your points below:
>
> **Weakness & Q1: Fairness performance on cross-dataset generalization**
> Thank you for raising this point, and we have updated our manuscript to include Rank Disparity in Table 2. We replicate that table here below. We note that the only dataset with usable protected attribute labels is AgeDB so thus, we present that result here. We divide the ages into groups of 0-25 yrs, 25-50yrs, 50-75yrs and 75-100yrs. Further, we report the maximum disparity amongst these groups. We note that the SMAC models are Pareto-*dominant* here showing the lowest error and lowest rank disparity.
> | Dataset  | Model       | Accuracy   | Disparity  |
> |----------|-------------|------------|------------|
> | CelebA   | DPN_CosFace | 64.84      | 0.2824     |
> |          | DPN_MagFace | 60.00      | 0.3129     |
> |          | SMAC_000    | 80.23      | 0.2188     |
> |          | SMAC_010    | **82.35**  | **0.1229** |
> | VGGFace2 | DPN_SGD     | 71.866     | 0.2247     |
> |          | DPN_AdamW   | 61.316     | 0.2114     |
> |          | Rexnet_100  | 59.1833    | 0.2892     |
> |          | SMAC_301    | **81.533** | **0.1883** |
>
> **Q2: Why are these architectures more fair?**
> Thank you for your question. We precisely search for a recurring Dual Path Network block in-terms of architecture. The handcrafted DPN block contains a Conv3x3Bn (Conv 3x3 followed by batchnorm), BnConv5x5 (batch norm followed by 5x5 convolution) and BnConv3x3 ( batch norm followed by 5x5 convolution). Amongst the architectures, we find a strong preference for the BnConv3x3 operation (every architecture containing at least one or more of such operations). Furthermore, in terms of the optimal face recognition head, we surprisingly find a strong preference for “CosFace” instead of “MagFace” and “ArcFace”. We find that “ArcFace” has the least preference during search. Moreover, we also discover that the SGD optimizer often with high learning rates > 0.1 is often preferred in comparison to AdamW and Adam optimizers from our search space.
>
> The proposed multi-objective neural architecture search and HPO simultaneously optimize two objectives, firstly the accuracy and secondly the fairness metric (e.g. rank disparity). Hence, we bias the search toward models which do not exploit the protected attribute (e.g. gender) to make classifications. We hypothesize that the SMAC models learn to use more fine grained facial features to distinguish faces instead of exploiting obvious coarse features like protected attributes (gender, race, age). We leave a more detailed analysis of the properties of learned features for future work.

---

> > ### Comment · Reviewer_3GhH · 2023-08-13
> >
> > I have read the authors' responses and reviewers' comments. The response addresses my concern. I raise my rating to "Strong Accept." I encourage the authors to add the response to the final version.

---

### Official Review · Reviewer_mXr9 · 2023-07-06

**Soundness:** 2 fair
**Presentation:** 3 good
**Contribution:** 4 excellent
**Rating:** 6
**Confidence:** 5

**Summary:**

This paper propose a brand new framework (NAS+HPO) to mitigate biases in FR. The discussion is extensive and interesting. But experiments on authoritative face recognition dataset are required, e.g., Ms1m, Glint360 and webface260m.

**Strengths:**

a. The presentation is easy to follow.
b. The discussion is extensive and interesting.
c. The paper propose a new framework (NAS+HPO jointly) to mitigate biases in FR.

**Weaknesses:**

The reported FR performance of this method should be also verified in large scale FR datasets, since lots of methods just work in small datasets but always fail in large ones. Some authoritative face recognition datasets are suggested, e.g., Ms1m v3[1], Glint360K[2] and webface260m[3].

[1] Lightweight face recognition challenge.
[2] Killing Two Birds with One Stone:Efficient and Robust Training of Face Recognition CNNs by Partial FC
[3] WebFace260M: A Benchmark Unveiling the Power of Million-Scale Deep Face Recognition

**Questions:**

null

**Limitations:**

null

---

> ### Author Rebuttal · Authors · 2023-08-09
>
> We’d like to first thank you for your time and thoughtful feedback on our manuscript. We appreciate that you find our presentation easy to follow, our discussion extensive and interesting. We have conducted new analysis and answer your question below:
>
> **New Results**
>
> **The Effect of Pretraining**
> We now study the effect of pre-training vs. training from scratch (Figure 2 (a) and (b) in rebuttal PDF) for face-recognition using the Dual Path Network architecture which is the basis for our search space definition. Interestingly, we find that the disparity of the pre-trained model is **much** higher compared to the model trained from scratch.  Moreover, we observe that while the pre-trained model starts strong in terms of accuracy, the model trained from scratch eventually catches up. This opens up an interesting direction of future work on how to effectively exploit pre-trained models for face-recognition systems without increasing bias.
>
> **SMAC Pareto-dominates other NAS Methods**
> We now study models discovered by other NAS methods (using a limited time-budget for search), and we observe that SMAC (multi-fidelity+Bayesian Optimization) optimizes compute-efficiency and performance.
>
> |                  |   Accuracy |   Rank Disparity |   Disparity |     Ratio |   Rank Ratio |   Error Ratio |
> |:-----------------|-----------:|-----------------:|------------:|----------:|-------------:|--------------:|
> | MO-ASHA_032 |   0.934739 |         0.390588 |   0.0485621 | 0.0533381 |     0.448144 |     0.542336 |
> | NSGA-II_728 |   0.868105 |         0.599085 |   0.0857516 | 0.103913  |     0.490213 |      **0.490651** |
> | SMAC_301    |  **0.963366** |         **0.230327** |   **0.0300871** | **0.0317269** |     **0.367554** |      0.582215 |
>
>
> **Fairness w.r.t. Age Groups on AgeDB**
> We conducted an analysis on fairness across age groups on AgeDB and find that our models are **pareto-dominant**.
>
> | Dataset  | Model       | Accuracy   | Disparity  |
> |----------|-------------|------------|------------|
> | CelebA   | DPN_CosFace | 64.84      | 0.2824     |
> |          | DPN_MagFace | 60.00      | 0.3129     |
> |          | SMAC_000    | 80.23      | 0.2188     |
> |          | SMAC_010    | **82.35**  | **0.1229** |
> | VGGFace2 | DPN_SGD     | 71.866     | 0.2247     |
> |          | DPN_AdamW   | 61.316     | 0.2114     |
> |          | Rexnet_100  | 59.1833    | 0.2892     |
> |          | SMAC_301    | **81.533** | **0.1883** |
>
>
> **Training on Larger Datasets**
> We appreciate your point that our learned architectures were not evaluated on additional very large-scale FR datasets. We did not conduct these experiments and leave them for future work since we specifically focused on datasets which have protected attribute labels, unlike Glint360K. During the rebuttal period, we were unable to obtain the WebFace360M dataset given the process and license agreement protocol. Finally, as the ethics reviewer has stated, the use of some of these datasets is controversial, MS-Celeb-1M for example is listed as a [deprecated dataset](https://neurips.cc/public/deprecated-datasets) by NeurIPS itself.

---

> > ### Comment · Reviewer_mXr9 · 2023-08-13
> > **S**
> >
> > some of my concern is resolved, I'll raise my score. good job.

---

### Official Review · Reviewer_Nw7v · 2023-07-06

**Soundness:** 4 excellent
**Presentation:** 4 excellent
**Contribution:** 4 excellent
**Rating:** 8
**Confidence:** 4

**Summary:**

The paper focuses on Bias Mitigation for face identification, i.e., ensuring that face identification works “well” for different identities: gender, race, etc. Unlike prior work, which focuses on model backbone agnostic methods to mitigate bias, this work explores the relevance of the inductive bias encoded in different Deep Learning model's backbones to the bias issue. In other words, are there model backbones better at learning robust features and ensuring good performance on samples from different identities? The authors uncover model configurations that significantly improve performance over standard model backbones through an extensive empirical analysis based on Neural Architecture Search on two face identification datasets. More surprisingly, the authors prove that these backbones' performance is better or more competitive than when paired with standard bias mitigation methods. Moreover, the authors confirm the generalizability of the configurations uncovered using CelebA and VGGFace2 by testing them on other datasets, further confirming their competitive performance. Finally, the authors analyze the new model configurations and ensure that the features learned by the models are less likely to be discriminative between the biased groups confirming that they learn more diverse and robust features.


**Strengths:**

1- The work is well motivated; prior work needs to include an analysis of model architecture relevance to the bias issue.

2-The results are interesting and relevant to fairness/bias community practitioners.

3-The advantages of newly discovered model configurations are explored by a well-designed empirical analysis that confirms the configurations' more robust learned features.


**Weaknesses:**

1- Some choices in the experimental design would benefit from further motivation. For example:
Why was the “multi-fidelity Bayesian optimization method SMAC3” chosen in particular? Are there other methods that could also work?


2- SMAC_301 was the architecture that works well across datasets. I understand that 301 denotes the operations that constitute the novel architecture. However,  the authors do not discuss the details of these operations or why they think they are meaningful choices compared to other choices ruled out by the NPS. Some discussion here would be helpful. Why do the authors think this configuration is better able to learn non-linearly separable bias features?

3- In Section 3, why are the models trained on a gender-balanced subset of the dataset? Wouldn’t one want to train on a gender-imbalanced split to see which architectures are less likely to be influenced by the imbalance? Is this the same in Section 4? Please clarify.

4- In the analysis in Section 4.2, were the hyper-parameters (learning rate, optimizer) of the timm models tuned too? Or were they the ones used by the original papers? If it is the latter, then it is an unfair comparison to the SMAC models since those hyper-parameters were tuned as outlined in Section 4.1.

5- From reading the paper in detail, I understand now that one of the motivations of Section 3 analysis was to limit the number of architectures explored in Section 4 (only DPN was considered since it achieved Pareto optimal performance on both datasets). However, I did not get that from the first read, so further clarification in the text about that would be helpful.

6- In Section 3, the hyper-parameters of the different models were not optimized. It is likely very computationally expensive. Nevertheless, it does undermine some elements of the analysis, so I would make that clear as limitations.


**Questions:**

I am overall positive about this work. However, I need further clarifications as outlined in the Weakness sections, particularly questions: (2,3,4). I am happy to increase my score upon adequate further clarification.


**Limitations:**

I commend the authors for explicitly discussing their work's technical limitations and that while it improves the notion of technical fairness, the advancement could still be harmful in downstream applications.

---

> ### Author Rebuttal · Authors · 2023-08-09
>
> Thank you for your time and thoughtful feedback on our manuscript. We appreciate that you see the novelty in our work being the first to systematically conduct a large-scale analysis on the problem of fairness face recognition with different architectures and hyperparameters. We address each of your points below.
>
> **W1: The choice of SMAC3**
> Thank you for your suggestion. We agree that given the plethora of methods for multi-objective NAS+HPO, there are multiple algorithms one could choose from. Given that SMAC3 supports parallelization across GPUs and multi-fidelity search, we initially restrict ourselves to SMAC3. Following your advice, we now studied two other multi-objective methods, MOASHA [1] and NSGA-II [2] from the syne-tune[3] library, using our search space design. Note that we run the search for a limited time budget of 48 hrs, so the models discovered would likely improve with a longer search budget. We will include an extended experiment in our updated manuscript.  We present the results below:
>
> |                  |   Accuracy |   Rank Disparity |   Disparity |     Ratio |   Rank Ratio |   Error Ratio |
> |:-----------------|-----------:|-----------------:|------------:|----------:|-------------:|--------------:|
> | MO-ASHA_032 |   0.934739 |         0.390588 |   0.0485621 | 0.0533381 |     0.448144 |     0.542336 |
> | NSGA-II_728 |   0.868105 |         0.599085 |   0.0857516 | 0.103913  |     0.490213 |      **0.490651** |
> | SMAC_301    |  **0.963366** |         **0.230327** |   **0.0300871** | **0.0317269** |     **0.367554** |      0.582215 |
>
>
> [1] Schmucker, R., Donini, M., Zafar, M.B., Salinas, D. and Archambeau, C., 2021. Multi-objective asynchronous successive halving. arXiv preprint arXiv:2106.12639
>
> [2] Deb, K., Pratap, A., Agarwal, S. and Meyarivan, T.A.M.T., 2002. A fast and elitist multiobjective genetic algorithm: NSGA-II. IEEE transactions on evolutionary computation, 6(2), pp.182-197
>
> [3] Salinas, D., Seeger, M., Klein, A., Perrone, V., Wistuba, M. and Archambeau, C., 2022, September. Syne tune: A library for large scale hyperparameter tuning and reproducible research. In International Conference on Automated Machine Learning (pp. 16-1). PMLR
>
>
>
> **W2: Why is SMAC_301 the best model?**
> In most of the fair models discovered by NAS+HPO, we see a prevalence of BnConv3x3 operation (every architecture containing at least one or more of such operations). Furthermore, in terms of the optimal face recognition head, we surprisingly find a strong preference for “CosFace” instead of “MagFace” and “ArcFace”. We find that “ArcFace” has the least preference during search. Moreover, we also discover that the SGD optimizer with high learning rates > 0.1 is often preferred in comparison to AdamW and Adam optimizers from our search space. We believe that these are just some of the important characteristics (architectural and hyper-parameter pipeline) for making models more fair.  We will include an extended discussion of these components in our updated manuscript.
>
> **W3: Gender-balanced training**
> We have employed the training regime for fair face identification as described by [4], which shows the importance of training models with fully balanced datasets (both balanced in identities and number of images per identity). They point out how these two types of imbalances (both at training and testing time) can cause researchers to draw misleading or incorrect conclusions. Thus, balancing the training and testing data as we did in our experiments is an important step to disaggregate the disparity introduced by the model architecture and hyperparameters, from the disparity introduced by the data imbalance.
>
> [4] Cherepanova, V., Reich, S., Dooley, S., Souri, H., Goldblum, M., & Goldstein, T. (2022). A deep dive into dataset imbalance and bias in face identification. Sixth AAAI/ACM Conference on Artifical Intelligence, Ethis, and Soceity, 2023.
>
>
> **W4: Hyperparameter tuning**
> We conduct our large scale analysis with handcrafted architectures and the hyperparameters as reported in their respective papers. In addition to this, we also study every model with 9-13 different hyperparameter combinations for each model, to allow for more flexibility in terms of optimizers, face-recognition heads, and learning rates (Section 3.2 Experimental Setup). Our goal is to compare these already strong pipelines with ones that can be discovered automatically using joint NAS+HPO.
>
> **W5&6: Clarity of writing**
> We greatly appreciate your careful read of our paper. We have updated the manuscript to incorporate this feedback, and we will include these edits in our updated manuscript.

---

> > ### Comment · Reviewer_Nw7v · 2023-08-13
> >
> > The response addresses my concerns. I encourage the authors to revise the manuscript and include the updates. I revised my score accordingly.

---

### Official Review · Reviewer_Kze1 · 2023-07-07

**Soundness:** 3 good
**Presentation:** 3 good
**Contribution:** 3 good
**Rating:** 6
**Confidence:** 4

**Summary:**

This paper presents a new perspective on bias mitigation in machine learning models, challenging the conventional belief that one should first find the highest-performing model and then apply a bias mitigation strategy. The authors propose that finding a fairer architecture offers significant gains compared to conventional bias mitigation strategies. To test this hypothesis, they conduct the first neural architecture search for fairness and a search for hyperparameters in face recognition.

**Strengths:**

This paper proposes a new way to mitigate biases in face recognition systems from the perspective of fairer model architectures.
This paper conducts the first large-scale analysis of the impact of architectures and hyperparameters on bias in face recognition, demonstrating that the implicit convention of choosing the highest-accuracy architectures is a sub-optimal strategy for fairness.
This paper may be the first to apply existing tools from NAS and HPO to design a fair face recognition model automatically.

**Weaknesses:**

（1）According to the definition of rank difference, the smaller the model's error, the fairer the model. Therefore, the difference in model parameters will lead to a difference in model performance. There is a lack of consistent constraints on the magnitude of the model parameters when searching for a fairer structure in this paper.
（2）Why the results in Table 2 are far worse than those reported in other papers.


**Questions:**

Please refer to the WeakNess

**Limitations:**

Yes

---

> ### Author Rebuttal · Authors · 2023-08-09
>
> We first thank you for your time and thoughtful feedback on our manuscript. We are glad that you find our approach novel and interesting. We address each of your questions below:
>
> **W1: Rank Disparity Definition**
> Thank you for raising this point. Precisely as per the definition of rank disparity, Rank(image) = 0 if and only if Error(image) = 0. This, however, **doesn’t** necessarily imply that decreasing error would correspond to decreasing rank disparity. Unlike the ratio of errors metric, rank disparity is a much richer metric which **doesn’t**  have a strong correlation with error rate.
>
> To probe this question, we conducted a new analysis (Figure 1 (a) in the PDF) which examines the correlation of each fairness-metric with model statistics. We compute statistics like number of parameters, model latency, number of convolutions, number of linear layers, and number of batch-norms in a model’s definition. Interestingly, we observe very low and non-significant correlations between parameter sizes and different fairness metrics. This observation supports the claim that increases in accuracy and decreases in disparity are very closely tied to the architectures and feature representations of the model, irrespective of the parameter size of the model. Hence, not constraining the parameter size helps our NAS+HPO approach search in a richer search space.
>
> **W2: Table 2 Results**
> Table 2 reports results of transfer learning from the given datasets (VGGFace2 on top and CelebA below) to the given datasets. Thus, the performance will be lower than if each model were fine-tuned or hyperparameters were optimized for each model on each dataset, or if a different pre-training dataset were used. We highlight here that the transfer learning result is strong and indicative that the representations that are learned by our novel architectures are indeed generalizable in a way that the other models are not. We have clarified this point in our updated manuscript.

---

> > ### Comment · Reviewer_Kze1 · 2023-08-14
> > **Response**
> >
> > I have read the authors' responses and reviewers' comments. The response addresses my concern. I keep my rating.

---

### Official Review · Reviewer_jfj2 · 2023-07-26

**Soundness:** 3 good
**Presentation:** 3 good
**Contribution:** 4 excellent
**Rating:** 7
**Confidence:** 4

**Summary:**

The authors offer a fresh view on mitigating fairness bias in ML by leveraging neural architecture (NAS) search and hyperparameter optimization (HPO).
The authors demonstrate their idea on the exemplary problem of face identification, where fairness biases have tangible consequences on society. They utilize a wide range of model architectures, and define a NAS+HPO search strategy where multi-objective optimization helps balance the tradeoff between accuracy and fairness (as quantified e.g. by the rank disparity metric) .


**Strengths:**

+ The authors formally describe a new paradigm compared with classical bias mitigation techniques that have traditionally focused on postprocessing/rectifying ML predictions, preprocessing/balancing the datasets, or extending the loss.
+ The solution is systematically devised and the experiments are straightforward to follow.
+ The experimental results are insightful and helpful in practice. I find it inspiring that the NAS models generalize to new protected attributes in new datasets.
+ The authors provide their source code besides a variety of analysis scenarios available to run via notebooks.

**Weaknesses:**

- There was no discussion on the impact of pretraining. With the availability of a large number of foundational models, it would be very relevant to shed light into which ones generalize better and why.
- The theoretical analysis is a. bit lacking. I was expecting more explanation of why the NAS models outperform other bias mitigation strategies and better generalize to other sensitive attributes. Is it because you are forcing the model to work harder and to avoid misusing these sensitive attributes as shortcuts when making predictions? (this would explain the reduced linear separability of protected attributes).
- The visualization of the results could be more insightful. For example a confusion matrix / similarity matrices might be useful to conduct error analysis and shed light into the improvements facilitated by the NAS paradigm. [Such analysis](https://arxiv.org/abs/2007.06068) has revealed many characteristics of VGGFace, e.g., oftentimes, gender misclassification is due to labeling issue (e.g. the image scrapped is not of the actor, but of their opposite-gender spouse).

Minor and language issues:
- The figures could be better annotated (e.g. to explain that the two red dots in Figure 2 are two variants of DPN)
- when comparing to other bias mitigation techniques => compared?
- at the most extreme low errors => unclear
- oepration
- to supports
- are Pareto-optimal the top performing … => are the Pareto-optimal top-performing …
- recognititon

**Questions:**

Have you considered other alternatives to SMAC3 or ParEGO? How generalizable are the insights in section 4.2. to possible alternatives?

**Limitations:**

Sufficiently discussed

---

> ### Author Rebuttal · Authors · 2023-08-09
>
> We thank you for your time and thoughtful feedback on our manuscript. We appreciate that you see our view on mitigating fairness bias in ML as fresh and interesting, our solution systematically devised, and our experiments straightforward to follow. Further, we are glad that you find our results insightful, inspiring and useful in practice. We address each of your questions below:
>
> **W1: The Effect of Pre-training**
> Thank you for raising this important point. Prompted by your feedback, we now fine-tuned a pre-trained DPN model obtained from timm on vggface-2, as DPN is the most representative of our search space. We compare the error trajectory of this model with and without pre-training. Interestingly, we observe that while the pre-trained model starts strong in terms of accuracy, the model trained from scratch catches up quickly. And more importantly, the disparity of the pre-trained model is **much** higher compared to the model trained from scratch. You can find the plot for the same in the attached PDF (Figure 2 (a) and (b)). We have updated our working draft accordingly, and we will perform more experiments to include in the updated manuscript.
>
> **W2: Theoretical Analysis**
> The proposed multi-objective neural architecture search and HPO simultaneously optimizes two objectives, firstly the accuracy and secondly the fairness metric (e.g. rank disparity). Hence, we bias the search toward models which do not exploit the protected attribute (e.g. gender) to make classifications. This is also reflected in the reduced linear separability of the features of the models discovered by SMAC. We hypothesize that the SMAC models learn to use more fine grained facial features to distinguish faces instead of exploiting obvious coarse features like protected attributes (gender, race, age). We leave a more detailed analysis on the properties of the features learned for future work.
>
> **W3: Visualizations**
> We appreciate this feedback. We have now conducted a new analysis in accordance with your suggestion in Figure 1 (a) of the attached PDF. Specifically, we visualize a dendrogram of the SMAC\_301 model as well as three highly performing DPNs. The visualization shows the correlation of the logits for each image in each identity. From this analysis, we observe that SMAC\_301 and DPN\_CosFace\_SGD have smaller cross-logit similarities, pointing to the fact that they do not try to exploit easier image properties like protected attributes to cluster images. The average similarities are lower for these models compared to others and the max similarities are much higher. We hypothesize that this property aids fair classifications.
>
> **Q1 SMAC3 and ParEGO Pareto-dominate other methods**
> We agree that given the plethora of methods for multi-objective NAS+HPO, there are multiple algorithms one could choose from. Given that SMAC3 supports parallelization and multi-fidelity search, we primarily restricted ourselves to it for compute optimal search. However, following your advice, we now studied two other multi-objective methods: MOASHA [1] and NSGA-II [2] from the syne-tune [3] library, using our search space design. Note that we run the search for a limited time budget of 48 hrs, so the models discovered may improve with a longer search budget. We will include an extended search comparison in our updated manuscript.  The results are found below and indicate that our chosen method Pareto-dominates the other methods for all metrics except for Error Ratio where it is Pareto-optimal.
>
> |                  |   Accuracy |   Rank Disparity |   Disparity |     Ratio |   Rank Ratio |   Error Ratio |
> |:-----------------|-----------:|-----------------:|------------:|----------:|-------------:|--------------:|
> | MO-ASHA_032 |   0.934739 |         0.390588 |   0.0485621 | 0.0533381 |     0.448144 |     0.542336 |
> | NSGA-II_728 |   0.868105 |         0.599085 |   0.0857516 | 0.103913  |     0.490213 |      **0.490651** |
> | SMAC_301    |  **0.963366** |         **0.230327** |   **0.0300871** | **0.0317269** |     **0.367554** |      0.582215 |
>
>
> [1] Schmucker, R., Donini, M., Zafar, M.B., Salinas, D. and Archambeau, C., 2021. Multi-objective asynchronous successive halving. arXiv preprint arXiv:2106.12639
>
> [2] Deb, K., Pratap, A., Agarwal, S. and Meyarivan, T.A.M.T., 2002. A fast and elitist multiobjective genetic algorithm: NSGA-II. IEEE transactions on evolutionary computation, 6(2), pp.182-197
>
> [3] Salinas, D., Seeger, M., Klein, A., Perrone, V., Wistuba, M. and Archambeau, C., 2022, September. Syne tune: A library for large scale hyperparameter tuning and reproducible research. In International Conference on Automated Machine Learning (pp. 16-1). PMLR

---

> > ### Comment · Reviewer_jfj2 · 2023-08-14
> >
> > I would like to thank the authors for their detailed response and for conducting the additional experiments. This confirmed my assessment about the utility and the solid results of the presented method.

---

### Author Rebuttal · Authors · 2023-08-09

We first thank all the reviewers for their insightful feedback and suggestions. Our work shows that bias in face recognition systems is actually inherent to their architectures and hyperparameters, and we can design fairer systems by searching for fair architectures, in fact significantly surpassing previous approaches. We appreciate that the reviewers find our perspective on bias mitigation interesting and fresh (**jfj2**, **kze1**, **Nw7v**, **3GhH**), our presentation clear and well-motivated (**jfj2**, **Nw7v**, **mXr9**, **3GhH**) and our experiments and analysis thorough and extensive ( **mXr9**, **3GhH**, **Nw7v**). Following suggestions made by the reviewers, we conducted further analyses and evaluations, some of which we highlight below:

**New Results**

1. We now conduct an analysis on fairness across age groups on AgeDB and find that our models are **Pareto-dominant**.

| Dataset  | Model       | Accuracy   | Disparity  |
|----------|-------------|------------|------------|
| CelebA   | DPN_CosFace | 64.84      | 0.2824     |
|          | DPN_MagFace | 60.00      | 0.3129     |
|          | SMAC_000    | 80.23      | 0.2188     |
|          | SMAC_010    | **82.35**  | **0.1229** |
| VGGFace2 | DPN_SGD     | 71.866     | 0.2247     |
|          | DPN_AdamW   | 61.316     | 0.2114     |
|          | Rexnet_100  | 59.1833    | 0.2892     |
|          | SMAC_301    | **81.533** | **0.1883** |

2. We now study models discovered by other NAS methods (using a limited time-budget for search), and we observe that SMAC (multi-fidelity+Bayesian Optimization) optimizes compute-efficiency and performance.

|                  |   Accuracy |   Rank Disparity |   Disparity |     Ratio |   Rank Ratio |   Error Ratio |
|:-----------------|-----------:|-----------------:|------------:|----------:|-------------:|--------------:|
| MO-ASHA_032 |   0.934739 |         0.390588 |   0.0485621 | 0.0533381 |     0.448144 |     0.542336 |
| NSGA-II_728 |   0.868105 |         0.599085 |   0.0857516 | 0.103913  |     0.490213 |      **0.490651** |
| SMAC_301    |  **0.963366** |         **0.230327** |   **0.0300871** | **0.0317269** |     **0.367554** |      0.582215 |


3. We now study the effect of pre-training vs. training from scratch (Figure 2 (a) and (b) in rebuttal PDF) for face-recognition using the Dual Path Network architecture which is the basis for our search space definition. Interestingly, we find that the disparity of the pre-trained model is **much** higher compared to the model trained from scratch. Moreover, we observe that while the pre-trained model starts strong in terms of accuracy, the model trained from scratch eventually catches up. This opens up an interesting direction of future work on how to effectively exploit pre-trained models for face-recognition systems without increasing bias.

**Ethical Concerns**
We strongly believe that our findings need to be placed into the larger sociotechnical context of facial recognition. The impacts of facial recognition technologies on individuals are well-documented, and our work considers a new way to reduce harms caused by disparities in these systems. We adhered to the [NeurIPS deprecated dataset guidelines](https://neurips.cc/public/deprecated-datasets) for our choice of datasets. MS-Celeb-1M and MegaFace are two datasets widely used by the face recognition community, even today, which we omitted from our experiments due to ethical issues. We have updated our manuscript to reflect these points and to further highlight the representational issues with CelebA as pointed out by reviewers.

---

### Decision · Program_Chairs · 2023-09-21

**Decision:**

Accept (oral)

**Comment:**

This paper addresses bias mitigation with Neural Architecture Search (NAS) and Hyperparameter Optimization (i.e. finding fairer architectures in contrast to finding a good performing architecture first and then applying bias mitigation strategies). The experiments focus on bias mitigation in face recognition and they are conducted on two popular benchmarks: CelebA and VGGFace2.

All the reviewers agreed on the novelty of the approach with respect to previous work on bias mitigation, and valued the extended experiments and the relevance of the results. The reviewers also highlighted some weaknesses and made some questions, which were satisfactorily addressed during the authors feedback (the authors provided new results, clarifications, and further discussions).

One reviewer flagged the paper for ethical revision, and the paper was reviewed by two ethics reviewers. One of the ethics reviewers found no ethical issues, but recommended extending the discussion on potential social and ethical impacts. The other ethics reviewer considered the paper to have ethical issues because of the use of the CelebA dataset. This dataset has representation bias and was collected without informed consent. The ethics reviewer recommends adding a discussion on the representation bias issue to the paper (the informed consent issue was already discussed in the ethics statement). The authors have already committed to follow these recommendations for the final version of the paper.

The paper was further discussed with the NeurIPS Ethics Chairs during the AC-SAC discussion period. During this discussion the Ethics Chairs request that the authors address these specific points for follow-up in the camera-ready manuscript:
1) To discuss if their test datasets (in Table 2, etc.) have similar ethical issues brought up
2) To discuss the limitations of their work, particularly as it might encourage techno-solutionism in the form of removing the bias and proceeding as usual without deeper considerations.